# The multiscale Routing Model mRM v1.0: simple river routing at resolutions from 1 to 50 km

Stephan Thober[1], Matthias Cuntz[2], Matthias Kelbling[1], Rohini Kumar[1], Juliane Mai[3], and Luis Samaniego[1]

[1]Computational Hydrosystems, Helmholtz Centre for Environmental Research - UFZ, Leipzig, Germany
[2]INRA, Université de Lorraine, AgroParisTech, UMR Silva, Nancy, France
[3]Civil and Environmental Engineering, University of Waterloo, Canada

**Correspondence:** Stephan Thober (stephan.thober@ufz.de)

**Abstract.** Routing streamflow through a river network is a fundamental requirement to verify lateral water fluxes simulated by hydrologic and land surface models. River routing is performed at diverse resolutions ranging from few kilometers to 1°. The presented multiscale Routing Model mRM calculates streamflow at diverse spatial and temporal resolutions. mRM solves the kinematic wave equation using a finite difference scheme. An adaptive time stepping scheme fulfilling a numerical stability criteria is introduced in this study and compared against the original parametrization of mRM that has been developed within the mesoscale Hydrologic Model (mHM). mRM requires a high-resolution river network, which is upscaled internally to the desired spatial resolution. The user can change the spatial resolution by simply changing a single number in the configuration file without any further adjustments of the input data. The performance of mRM is investigated on two datasets: a high-resolution German dataset and a slightly lower resolved European dataset. The adaptive time stepping scheme within mRM shows a remarkable scalability compared to its predecessor. Median Kling-Gupta efficiencies change less than 3 percent when the model parametrization is transferred from 3 to 48 km resolution. mRM also exhibits seamless scalability in time, providing similar results when forced with hourly and daily runoff. The streamflow calculated over the Danube catchment by the Regional Climate Model REMO coupled to mRM reveals that the 50 km simulation shows a smaller bias with respect to observations than the simulation at 12 km resolution. The mRM source code is freely available and highly modular, facilitating easy internal coupling in existing Earth System Models.

## 1 Introduction

Streamflow provides an integrated signal of lateral hydrologic fluxes at the land surface over a catchment area. Streamflow observations are routinely used within hydrologic modeling to perform model characterization or calibration/validation (Beven, 2012). Comparisons between simulated and observed streamflow are typically conducted using measures focusing on daily values (Nash and Sutcliffe, 1970; Gupta et al., 2009). Similarly to hydrologic models (HM), land-surface models (LSM) also represent the terrestrial hydrologic cycle. They additionally include the terrestrial energy budget and biogeochemical cycles, such as the carbon cycle, to provide the exchange fluxes of the land-surface with the atmosphere in, for example, Regional Circulation Models (RCM) or Earth System Models (ESM). Streamflow is estimated in ESMs to provide fresh water input of

the land surface into the ocean (Sein et al., 2015). Streamflow observations are also used in land surface models in the context of climate studies to validate the hydrologic cycle at daily (a.o., Marx et al., 2018; Thober et al., 2018; Samaniego et al., 2018), monthly (a.o., Hagemann et al., 2009; Li et al., 2015; Zhang et al., 2016) or even annual time scales (a.o., Zhou et al., 2012).

Hydrologic models and land surface models (LSM) are historically run at different spatial resolutions. Hydrologic models
(HM) are, for instance, applied at scales of few kilometers and less, even in continental-scale applications (Wood et al., 2011; Marx et al., 2018; Thober et al., 2018; Samaniego et al., 2018; Wanders et al., 2019), whereas LSMs are applied at resolutions of tens of kilometers and more within climate change studies (Van der Linden and Mitchell, 2009; Taylor et al., 2012). However, a substantial increase in model resolution has been achieved for Regional Circulation Models (RCM) in the past years and these are run nowadays at different resolutions down to the convection-permitting scale of a few kilometers (Jacob et al.,
2014). Applying RCMs at diverse resolutions implies that the same LSM (i.e., the same representation of water, energy, and biogeochemical processes) is used on diverse resolutions. This imposes a challenge on the LSM parametrizations to be able to represent all included processes at the different resolutions (Wood et al., 1998; Haddeland et al., 2002; Boone et al., 2004). The main goal of this study is to provide LSMs, that do not include a river routing scheme, with a framework to compute streamflow for comparison against observations. The distinctive property of this framework is that the spatial resolution can be
easily changed by the user without any modification of the model setup.

River routing is the process of predicting the hydrograph evolution as runoff moves through a river network. It can be described at different levels of complexity. The general governing equation of this phenomena are the uni-dimensional Saint-Venant equations (de Saint-Venant, 1871). Models using the Saint-Venant equations are referred to as hydraulic models that are especially suited if backwater effects occur such as in flat regions or river deltas (Paiva et al., 2011; Miguez Macho and
Fan, 2012; Paiva et al., 2013; Yamazaki et al., 2013). These models exploit remote sensing to derive model parameters and setup (Neal et al., 2012; Beighley et al., 2009). If rivers are steep enough and relatively shallow, simplification of the Saint-Venant equations such as the kinematic wave equation are sufficient (Lohmann et al., 1996; Hagemann and Dümenil, 1997; Todini, 2007). They require less information about river topography and only account for wave advection and attenuation. These models are not applicable for large river basins with extensive floodplains such as the Amazonas and Niger because
they cannot account for floodplain inundation (Neal et al., 2012; Getirana et al., 2012; Pontes et al., 2017), which causes a negatively skewed hydrograph (Collischonn et al., 2017). It is worth noting that the impact of floodplain processes dominates the differences between a hydraulic model and kinematic wave models (Paiva et al., 2013).

A common approach to achieve scale-independent streamflow is to perform the river routing calculations on a fixed spatial and temporal resolution, regardless of the resolution of the hydrologic or land surface model providing the input runoff flux.
Global routing schemes, for example, often use fixed 0.5° or 1.0° resolutions to produce river discharge of large river basins globally (Hagemann and Dümenil, 1997; Oki et al., 1999; Pappenberger et al., 2009; Hagemann et al., 2009). Within hydrologic models, high-resolution routing algorithms at fixed scales of 5 to 16 km are used (David et al., 2011; Kumar et al., 2013b, a). Only few studies have explicitly investigated the spatial scaling capabilities of existing routing approaches by introducing sub-grid and between-grid heterogeneities (Li et al., 2013).

The main objective of the multiscale Routing Model mRM presented in this study is to provide a simple in both, model complexity and applicability, river routing of hydrologic and land surface model outputs at various spatial resolutions ranging from the kilometer scale to large scales at 50 km or more in a seamless way (Samaniego et al., 2017b). The stand-alone model allows the user to adjust freely the spatial resolution by simply changing a single value in a configuration file without any further modifications of the input data. The model resolution should thereby not influence the obtained streamflow, otherwise model re-calibrations at each resolution would be necessary. mRM also keeps the computational demand to a minimum (see Appendix B for details on run times of mRM), one major advantage of a scalable modeling system (Kumar et al., 2013a).

The analysis of the scaling capabilities of the multiscale Routing Model mRM is shown for 622 European catchments ranging from 100 km2 to 100,000 km2 in size and spatial resolutions from 1 to 48 km (section 3.2). The river network has to be provided only at the finest spatial resolution supported by the available data, for example a digital elevation map. This high-resolution river network is then internally upscaled to the resolution specified by the user in a configuration file. The upscaling accounts for the correct representation of the catchment area/stream network without any further data requirement (section 2.4). A parameter sensitivity analysis is presented for the 622 catchments, which highlights the small influence of the model parameter of mRM (section 3.1). The multiscale Routing Model mRM is coupled internally to the mesoscale Hydrologic Model mHM (Samaniego et al., 2010; Kumar et al., 2013b) and the improvement of mRM over the original routing parametrization in mHM is demonstrated (section 3.3). The overall focus of mRM is to provide a simple routing tool that can be coupled to any land-surface and hydrologic model across several spatial resolutions, and allowing them to access streamflow observations. mRM is applied as a stand-alone post-processor to the output of the REMO Regional Climate model over the Danube catchment for demonstration (section 3.4).

## 2  Description of the multiscale Routing Model mRM

### 2.1  Finite Difference Approximation of Kinematic Wave Equation

The multiscale Routing Model mRM uses the kinematic wave equation, first analysed by Lighthill and Whitham (1955), to describe the water flow within a stream as

$$\frac{\partial Q}{\partial t} + c\frac{\partial Q}{\partial x} = 0,$$  (1)

where $Q$ (m$^3$ s$^{-1}$) is streamflow, $x$ (m) the space dimension, $t$ (s) the time dimension, and $c$ (m s$^{-1}$) the celerity. The kinematic wave equation is a simplification of the Saint-Venant equations (Chow et al., 1988). The derivation is based on the assumption that the continuity equation is sufficient to describe the movement of flood waves. In detail, a constant river bed slope and time-invariant celerity $c$ have to be assumed (Lighthill and Whitham, 1955). Kinematic waves account for the advection of water but not for complex fluvial processes such as flood wave attenuation, backwater effects, and floodplain inundation. It is however widely used because advection is the governing fluvial process as long as backwater and floodplain inundation effects can be neglected (Paiva et al., 2013). mRM employs the classical finite difference weighted approximation on a four point scheme to

solve equation (1). Details about the derivation can be found in Chow et al. (1988) and Todini (2007). It is summarized shortly in the following.

The partial derivatives within equation (1) are represented as finite differences between four values, that means on two locations at two points in time:

$$\frac{\partial Q}{\partial t} \approx \frac{\epsilon(Q(x_j,t_{i+1})-Q(x_j,t_i))+(1-\epsilon)(Q(x_{j+1},t_{i+1})-Q(x_{j+1},t_i))}{\Delta t}, \tag{2}$$

$$\frac{\partial Q}{\partial x} \approx \frac{\varphi(Q(x_{j+1},t_{i+1})-Q(x_j,t_{i+1}))+(1-\varphi)(Q(x_{j+1},t_i)-Q(x_j,t_i))}{\Delta x},$$

where $j$ denotes the spatial location (i.e., reach id) and $i$ denotes the timestep. $\epsilon$ is a space-weighting factor and $\varphi$ is a time-weighting factor. mRM uses a rectangular grid to represent the river network with a river reach in the model connecting two grid center locations. Different spatial locations are separated by $\Delta x$ and time steps by $\Delta t$. The two weighting factors, $\epsilon$ and $\varphi$, can

be chosen between 0 and 1, but the numerical solution becomes unstable for $\epsilon > 0.5$ (Cunge, 1969). The numerical diffusion depends linearly on $\epsilon$ (Cunge, 1969), with 0 implying full numerical diffusion and 0.5 no numerical diffusion, respectively.

Setting $\varphi$ to 0.5, which represents a time-centered scheme, and substituting equation (2) into equation (1) results in the classical linear equation:

$$Q(x_{j+1},t_{i+1}) = C_1 Q(x_j,t_{i+1}) + C_2 Q(x_j,t_i) + C_3 Q(x_{j+1},t_i), \tag{3}$$

with the coefficients $C_1$, $C_2$ and $C_3$ being:

$$C_1 = \frac{-2\Delta x\epsilon + c\Delta t}{2\Delta x(1-\epsilon)+c\Delta t},$$

$$C_2 = \frac{2\Delta x\epsilon + c\Delta t}{2\Delta x(1-\epsilon)+c\Delta t}, \tag{4}$$

$$C_3 = \frac{2\Delta x(1-\epsilon) - c\Delta t}{2\Delta x(1-\epsilon)+c\Delta t}.$$

The coefficients $C_1$, $C_2$ and $C_3$ add up to one. The spatial resolution applied in equation (3) is called "routing" resolution in

the following.

## 2.2 Stream Celerity Parametrization based on Terrain Slope

Two parametrizations of equation (4) are available in mRM: first, the regionalized Muskingum-Cunge (rMC) parametrization with a fixed time step as implemented in the mesoscale Hydrologic Model mHM presented in Samaniego et al. (2010) and Kumar et al. (2013b), and second, the newly developed parametrization using spatially varying celerities in combination with

an adaptive time step (aTS). A short summary of the former is presented in the Appendix A and is referred to as rMC in the following. The latter is described in detail in this and the next section and is referred to as aTS scheme.

The aTS calculates stream celerity as a function of terrain slope using a simple relationship:

$$c_i = \gamma\sqrt{s_i}, \tag{5}$$

where $c_i$, $s_i$ and $\gamma$ are celerity, terrain slope and a free model parameter, respectively, and $i$ is the grid cell index. Equation (5)

was proposed by Miller et al. (1994) for evaluating the accuracy of atmospheric GCMs against streamflow observations. They

used $\gamma = 49$ with a topography at 5' resolution (ca. 10 km at the equator). Coe (2000) used the same formulation also at 5' resolution but set $\gamma = 113$. $\gamma = c_0/\sqrt{s_0}$ is the ratio of a minimum celerity $c_0$ over the square root of a reference slope $s_0$. The latter should depend on the resolution of the input data so that the aTS model parameter $\gamma$ should theoretically also depend on the resolution of the underlying terrain data, i.e., the digital elevation model (DEM). Because both parameters, $c_0$ and $s_0$, are unknown, aTS conceptualizes them as one effective parameter $\gamma$. It is set to range between 0.1 and 30 in this study because values above 30 led to unrealistic celerities with the two used DEMs of 100 m and 500 m resolution (see below). The parametrization used here (equation 5) is an alternative to Manning's equation (Manning, 1891; Chow et al., 1988), which is more physically based than equation (5), but additionally requires information of river cross sections and Manning's roughness coefficient, which need to be parametrized if observations are not available. Manning's equation thus typically requires more parameters than equation (5). The simplified representation used in aTS because of its sufficiently high model performance and its simplicity (see Section 3.1).

The celerity relationship (equation 5) is applied at the resolution of the digital elevation model (DEM), from which terrain slope is derived. Ideally, channel slope instead of terrain slope should be used in equation (5), but it is not as readily available as terrain slope. Applying equation (5) at the resolution of the DEM provides a compromise because a high-resolution DEM provides a close approximation of channel slope. A median absolute deviation (MAD) filter (Sachs, 1999) is applied to the high resolution slope data along the path of the main river with a threshold value of 2.25 to correct for outliers. Large slope outliers happen easily in DEMs, for example, when the river flows in a valley and one grid cell represents the valley while the next grid cell represents the hill top. A minimum river bed slope of 0.1% is further assumed to avoid numerical instabilities in flat terrains. The celerities are then upscaled to the routing resolution, i.e., the resolution at which the kinematic wave equation is solved (equation 3). The upscaling is by averaging with the harmonic mean, the correct averaging operator for celerities (or speed). This follows the Multiscale Parameter Regionalization (MPR) approach (Samaniego et al., 2010; Kumar et al., 2013b), which relates model parameters to physiographic characteristics at the highest possible resolution. The upscaling considers also only those high-resolution grid cells that align with the main river network because aTS only considers the flow in the main river reach, assuming that travel times in the main reach dominate the routing process in tributaries. Alternative models such as MOSART (Li et al., 2013) also consider flow in tributaries and head waters.

## 2.3 Adaptive Time Step (aTS) Implementation

The aTS scheme uses an adaptive time step to calculate the linear coefficients in equation (4). The basic idea is that the time step should be such that water has not been transported further than $\Delta x$ during a single step. This condition is generally known as Courant-Friedrich-Lewy criterium, which is a necessary condition for numerical stability of finite difference schemes (Courant et al., 1928). The condition can be expressed as:

$$C_r = \frac{c\Delta t}{\Delta x} \leq C_{max} = 1, \tag{6}$$

where $C_{max}$ is the Courant number. aTS uses a Courant number of 1 (Bates et al., 2010). The Courant condition couples the applied spatial resolution with the integration time step of the finite difference scheme. Celerities $c_i$ are typically in the order

of a few $\mathrm{m\,s^{-1}}$, calculated using equation (5) and averaged harmonically along the river path. Spatial grids are in the range of a few kilometers to around $100\,\mathrm{km}$ in the case of regional to continental applications. The time step $\Delta t$ is chosen such that it does not deviate too much from the Courant number $C_{max}$ (equation 6) to keep computational demand to a minimum (see Appendix B for details on run times of mRM). This implies that $\Delta t$ ranges from a few minutes for high-resolution grids to a

few hours for continental scale applications.

In detail, aTS chooses $\Delta t$ from a set of prescribed values such that $c_i \Delta t / \Delta x$ is close to but less than 1 for all celerities $c_i$. The prescribed values range from one minute to one day, namely: $1\,\mathrm{min}$, $2\,\mathrm{min}$, $3\,\mathrm{min}$, $4\,\mathrm{min}$, $5\,\mathrm{min}$, $6\,\mathrm{min}$, $10\,\mathrm{min}$, $12\,\mathrm{min}$, $15\,\mathrm{min}$, $20\,\mathrm{min}$, $30\,\mathrm{min}$, $1\,\mathrm{h}$, $2\,\mathrm{h}$, $3\,\mathrm{h}$, $4\,\mathrm{h}$, $6\,\mathrm{h}$, $8\,\mathrm{h}$, $12\,\mathrm{h}$, and $1\,\mathrm{day}$. The choice of these values is motivated from the fact that these represent multiples or fractions of hourly and daily time steps. These time steps allow in principle model applications

from $100\,\mathrm{m}$ to $100\,\mathrm{km}$, for typical celerities around $1.5\,\mathrm{m\,s^{-1}}$.

Note that the chosen time step depends only on the spatial resolution and is independent of the time resolution of the provided forcing. For example, applying aTS at $12\,\mathrm{km}$ resolution using a celerity of $c = 1.5\,\mathrm{m\,s^{-1}}$ gives $\Delta x / c$ of $2.2\,\mathrm{hours}$ and, hence, a time step of two hours will be chosen. If aTS is forced with hourly input, it will aggregate the input over two consecutive time steps prior to the routing. The calculated streamflow is then distributed to the previous two time steps because these represent

the mean flow over this period. If aTS is forced with daily input, it will use internally 12 iterations of 2-hour time steps to route the water through the river network. In this case, aTS will also return the average of the calculated 12 streamflow values at each reach.

## 2.4 Data Requirements and Model Setup

Three different input sources are required to run the multiscale Routing Model mRM. First, mRM requires information about

the river network. mRM uses a rectangular grid to represent the river network over a domain (Yamazaki et al., 2011). Water can only be transported from a specific grid cell to one of the eight neighboring cells following the steepest slope direction (D8 method, O'Calaghan and Mark, 1984). This procedure has to be carried at the highest possible resolution supported by the available dataset. For example, a high-resolution $100\,\mathrm{m}$ digital elevation model (DEM) can be used to calculate flow directions following the D8 method (Figure 1 top left). It is worth mentioning that DEMs typically have to be adjusted to align with

existing river networks using additional information about river locations (Döll and Lehner, 2002). High-resolution datasets such as HydroSHEDS (Lehner et al., 2008) can be used alternatively. Once the high-resolution flow direction is obtained following the nomenclature of 1 – east to 128 – north east clockwise, it is internally upscaled in mRM (Figure 1 center top) to the routing resolution specified by the user, employing the method of Döll and Lehner (2002). This upscaling technique has already been implemented in Samaniego et al. (2010). The flow direction at a low resolution grid cell ($3 \times 3$ grid in Figure 1) is

equal to the flow direction of the underlying high resolution grid cell with the highest flow accumulation. If this high resolution grid cell is not on an edge of the low resolution grid cell, then the low resolution grid cell is an outflow of the domain. It is worth mentioning that this procedure leads to incorrect basin areas at coarse resolution (Yamazaki et al., 2009). A detailed investigation of four large European river basins reveals that basin area is correctly calculated for model resolutions less than

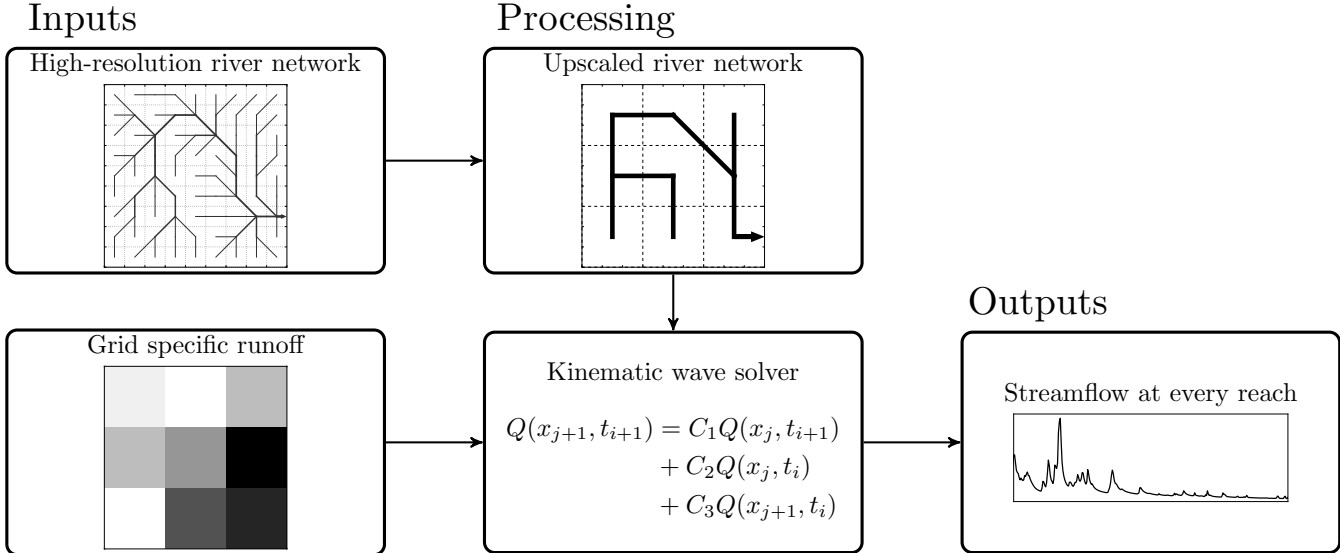

**Figure 1.** Flowchart of the processing steps required to run the multiscale Routing Model mRM: left, input: static high-resolution information about the routing network and dynamic runoff field; middle, internal mRM processing: aggregation of static and dynamic data to the routing resolution given by the user and execution of routing at this resolution; right, output: calculated streamflow at prescribed gauge.

km (see Appendix C). At these resolutions, mRM will route water to the correct ocean basin in global scale applications. Note that mRM can also handle rotated grids, if the high-resolution digital elevation model is provided on a rotated grid.

Second, the gridded runoff fields have to be provided in Network Common Data Form (NetCDF). Units of the forcing can be either $mm\,h^{-1}$ or $kg\,m^{-2}\,s^{-1}$ to facilitate applications to hydrologic models as well as land surface models. The most common use case is that mRM is applied to the sum of all runoff components of the driving model, which is based on the assumption that all components enter the river in the same grid cell. However, it is possible to apply mRM to different components individually which can be used as a model diagnostic. We acknowledge that equation (1) considers the entire channel flow. Its application to individual runoff components should be interpreted with caution, which may require conceptualizations for different flow celerities and varying topography among other factors. The spatial resolution of the runoff field is required to be a multiple of the resolution of the flow direction field. The most common use case is that streamflow is calculated at the resolution of the runoff and mRM will upscale the river network to the resolution as described before. However, mRM puts no constraints on model resolution. Simulations at higher or lower resolutions can be conducted as long as it is a multiple of both, the runoff grid and the grid of the flow directions. In this case, runoff will be up-/down-scaled employing weighted area fractions, which guarantees mass conservation. When mRM is coupled to coarse scale simulations (e.g., spatial resolution of $1°$), it is advisable to choose a lower resolution for mRM to correctly represent basin areas (see Appendix C).

Third, observed river streamflow can be provided to mRM at multiple locations within the river network. These locations have to be specified within the high-resolution river network and are then located on the upscaled river network internally

within mRM. However, users should be cautious when selecting the model resolution so that the streams represented by the gauging data are still resolved within the upscaled river network. Thus, the upscaled flow accumulation in each grid cell is given in an output NetCDF file, which allows comparison to the drainage area of a given gauge. Observed discharge data is not required for mRM when applied, for example, at ungauged locations. It is, however, mandatory when performing model optimization. Different error measures such as Nash-Sutcliffe efficiency (Nash and Sutcliffe, 1970) and Kling-Gupta efficiency (Gupta et al., 2009) can be calculated directly in mRM to inform the user about model performance.

A test basin is provided alongside the model code to illustrate the different data required to run the model and their formatting. The model code also contains pre-processing scripts to calculate the flow direction from a given DEM, or flow accumulation from given flow directions.

## 2.5 Experimental Setup

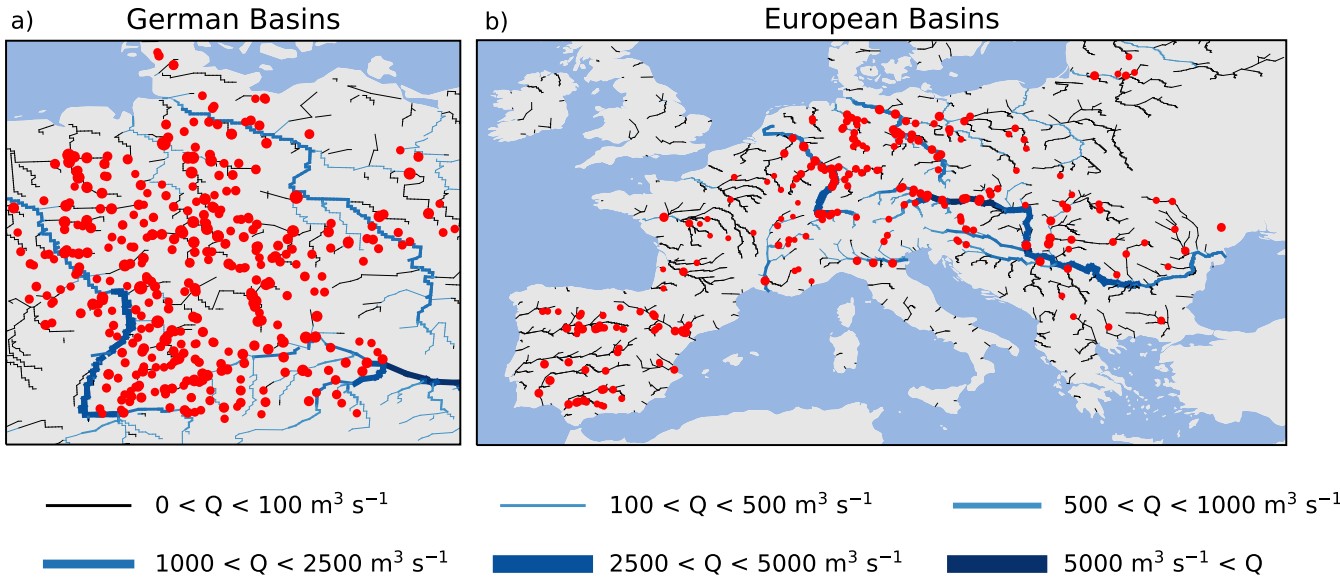

**Figure 2.** Discharge gauges used for the evaluation of the multiscale Routing Model mRM: a) 368 gauges in the German dataset, b) 254 gauges in the European dataset. In the background, the results of a pan-European simulation using the multiscale Routing Model mRM and the mesoscale Hydrologic Model mHM at a 5 km resolution is shown. The simulated streamflow Q is depicted for the 5th Aug 2002.

A total of 622 stream gauges are used in this study to assess the performance and scaling capabilities of mRM (Figure 2). These contain 368 basins in the German dataset (Figure 2a) and 254 basins in the European dataset (Figure 2b). Input for mRM is derived from simulations carried out with the mesoscale Hydrologic Model mHM (Samaniego et al., 2010; Kumar et al., 2013b). Two different model setups for mHM are used in this study. The setup for the German dataset is identical to Samaniego et al. (2013) and Zink et al. (2016) with details given in Zink et al. (2017). The flow direction and accumulation are derived from a 100 m DEM. The setup for the European dataset was used in Thober et al. (2015) with details given in Rakovec

et al. (2016). The DEM used to derive the river network has a 500 m resolution in this case. Runoff simulated by mHM was stored at hourly and daily resolution in NetCDF files for both sets of basins. mHM simulations for the German dataset are provided at 4 km resolution while European simulations are provided at 24 km spatial resolution. The difference originates from the available meteorological forcing datasets, which are derived from station data of the German weather service at 4 km resolution (Zink et al., 2017) for the German dataset while E-OBS data at 24 km resolution (Haylock et al., 2008) was used for the European dataset. To study the spatial scalability of mRM, streamflow is routed at different spatial resolutions, which are 1 km, 2 km, 4 km, 8 km, and 16 km for the German dataset and 3 km, 6 km, 12 km, 24 km, and 48 km for the European dataset. The selected resolutions cover a range of hydrologic applications from small to large scale modeling studies (Wood et al., 2011; Samaniego et al., 2017a) as well as scales of 0.5° used in climate studies (Taylor et al., 2012; Jacob et al., 2014). Input runoff on 4 km for the German dataset and on 24 km for the European dataset is rescaled internally in mRM to the desired routing resolution that is provided in a configuration file.

The mesoscale Hydrologic Model mHM coupled to mRM using the regionalized Muskingum-Cunge (rMC) and adaptive time step (aTS) parametrization were calibrated at all catchments to provide a realistic representation of the hydrologic cycle. Details about model calibration can be found in Samaniego et al. (2010); Kumar et al. (2013b); Rakovec et al. (2016), and Zink et al. (2017). The calibrations of both mHM and mRM parameters are carried out using the Shuffled Complex Evolution (SCE) algorithm (Duan et al., 1992). SCE is coupled internally to both models and SCE parameters (e.g., number of complexes) can be specified by users in a namelist.

The Kling-Gupta efficiency (KGE) is selected as a metric for evaluating model performance (Gupta et al., 2009). KGE is composed of three measures that relate simulated and observed streamflow. These are the ratio of simulated and observed mean values, the ratio of simulated and observed standard deviations, and the Pearson correlation coefficient. In comparison to the Nash-Sutcliffe efficiency (NSE, Nash and Sutcliffe, 1970), KGE provides a more balanced metric that is less sensitive to high streamflow values than NSE.

The model calibration and evaluation employs daily values of observed streamflow. The observed values are obtained from the Global Runoff Data Centre (GRDC) for the period from 1950 to 2010. Although mRM is run internally at higher temporal resolutions, the simulated streamflow is eventually aggregated to daily values for comparison against observations. Daily observed values are chosen here because the hydrologic cycle can be investigated in greater detail compared to monthly values commonly used with land surface models (e.g., Hagemann et al., 2009; Zhang et al., 2016).

## 3 Results

### 3.1 General Model Performance and Parameter Sensitivities

The adaptive time step parameterization (aTS) in mRM contains one parameter representing the relationship between terrain slope and streamflow celerity ($\gamma$ in equation 5). There is also an adjustable coefficient for the space weighting in the finite difference solver ($\epsilon$ in equation 2). The sensitivity of aTS to $\epsilon$ and $\gamma$ is explored here. The performance of simulated streamflow of aTS appears to be very high in general almost independent of the choice of $\epsilon$ and $\gamma$.

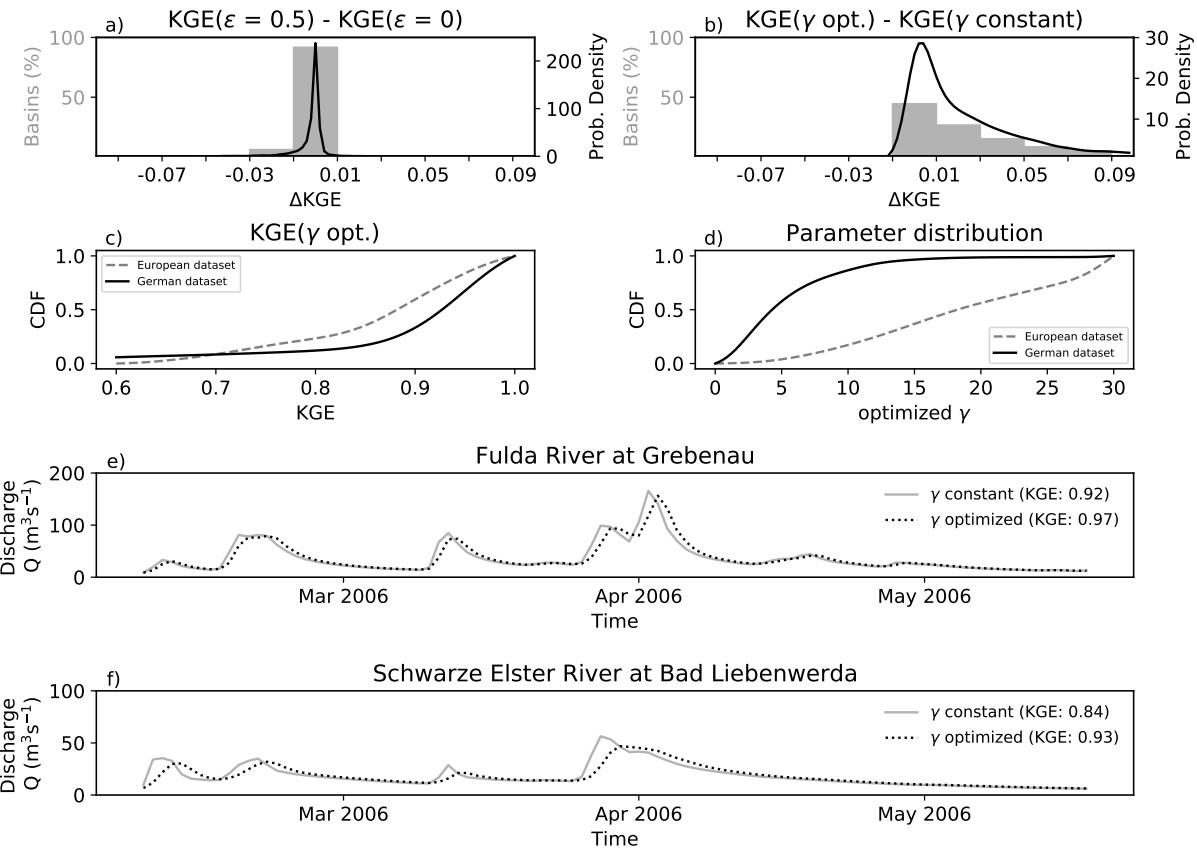

**Figure 3.** Differences in KGE between no numeric diffusion ($\epsilon = 0.5$) and full numeric diffusion ($\epsilon = 0.$) for the combined European and German datasets (panel a), and differences in KGE between model runs with optimized parameter values at each gauge and one constant parameter for all gauges (panel b). Probability density functions (PDF) are shown as black lines. The integrals of the PDFs over intervals of 0.02 (e.g., -0.01 to 0.01) are shown as gray bars normalized with respect to all basins. Panel c: Cumulative distribution functions (cdfs) of Kling-Gupta Efficiencies (KGE) for the European and German datasets separately based on optimized parameters $\gamma$. Panel d: CDF of optimized $\gamma$ for the two datasets. The underlying data shown in panels a-d is pooled for all catchments and all resolutions. Panels e) and f) show the hydrographs for two catchments at 4 km resolution for a parameter value of $\gamma = 15$ (solid gray line) and the optimized value (dashed black line).

The density function peaks around $\Delta$KGE = 0 for the space-weighting factor $\epsilon$ (Figure 3a). The $\Delta$KGE estimated across the investigated basins are within the interval $-0.03$ to $0.01$. All changes in KGE below a magnitude of $0.01$ are considered negligible, in alignment with previous literature, corresponding roughly to an error level in streamflow of $1\,\mathrm{mm\,d^{-1}}$ (Kollat et al., 2012). Some large basins in the European dataset show up to 0.03 higher KGE values using $\epsilon = 0$ compared to $\epsilon = 0.5$.

Note that the numerical diffusion of this finite difference solver (equation 4) depends linearly on $\epsilon$ (Cunge, 1969). An $\epsilon$ value of 0 corresponds to full numeric diffusion, whereas a value of 0.5 to no diffusion. The numerical diffusion is often chosen to correspond to the actual physical diffusion of the river by adjusting the value of $\epsilon$ (Todini, 2007; Beighley et al., 2009). aTS is using a space-weighting factor $\epsilon = 0$ because this value provides slightly better estimates than a value of 0.5, but the impact of
this factor is overall negligible.

The density function of $\Delta$KGE is skewed when comparing the performances between optimized $\gamma$ values at each gauging station and resolution with a constant value of 15 for all stations (Figure 3b). A value of 15 is chosen because it provides the best compromise solution of the obtained optimized values (Figure 3d). It can be expected that optimized parameters give higher performances than a fixed value. The performance increments with optimized parameters were, however, less than 0.01
in more than 37% of the basins while only about 42% of the basins exhibited $\Delta$KGE values of 0.01 to 0.05. This means that performance increments with optimized $\gamma$ values were below 0.05 in 79% of the basins (Figure 3b).

Overall, the KGE values for the European and German dataset are very high with only 4% of the basins exhibiting a KGE value less than 0.6 (Figure 3c). The median KGE values are 0.89 and 0.94 for the European and German dataset, respectively. KGE values are, however, highly dependent on the used hydrologic model (i.e., mHM) and the quality of the input data.
The hydrologic model determines the partitioning of precipitation into evapotranspiration and runoff as well as the temporal dynamics of generated runoff. It thus affects all three components of KGE: the bias, the variance and the correlation. The routing model, on the other hand, is not able to change the long-term water balance and is thus not affecting the bias term of the KGE. The routing model is, however, able to change the dynamics of simulated streamflow and thus greatly affect the variance term of KGE. The distribution of the optimized parameter values is very different for the German and the European datasets
with median values of 4 and 21, respectively (Figure 3d). These differences originate from the resolution of the underlying digital elevation model (DEM) and hence the slopes used in equation (5). The slope data for the German dataset is available at a 100 m resolution, while it is at 500 m resolution for the European dataset. The slopes will be larger and more variable at 100 m resolution compared to 500 m resolution. This implies that lower slope values (European dataset) are associated with higher $\gamma$ values and higher slope values (German dataset) are associated with lower $\gamma$ values, which results in similar celerities
for the two datasets. This highlights that the obtained parameter values are highly dependent on the underlying dataset, which has been identified as a major source of hydrologic modelling uncertainty (Livneh et al., 2015).

Hydrographs for two German river basins that exhibit $\Delta$KGEs of 0.05 and 0.11, respectively, are shown in Figure 3e and Figure 3f. These $\Delta$KGE values were among the highest of all basins and model resolutions considered in this study. A shift in peak flows of about one day can be spotted visually at $\Delta$KGE values of 0.05 (Figure 3e). This difference is representative
for around 21% of all catchments. A difference in KGE of 0.11 implies a change in the amount and timing of peak flows (Figure 3f) and is representative for around 8% of all catchments. The overall recession dynamics are comparable independent of the change in $\gamma$ (Figure 3e and 3f). Moreover, no substantial shift in amount and timing of peak flows is observed in 79% of the catchments. It will ultimately depend on the preference of the model user if parameter calibration is applied for a specific use case.

### 3.2 Temporal and Spatial Scalability

The aTS scheme is run first with different temporally aggregated inputs and second on different spatial resolutions to demonstrate its scalability across time and space.

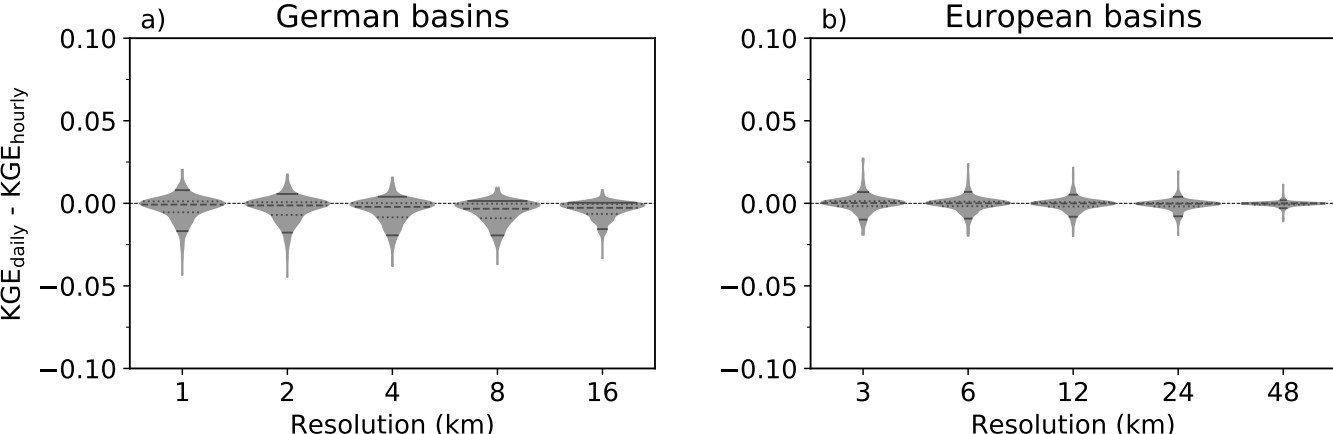

**Figure 4.** Probability density functions (PDF) of differences in KGE values obtained with hourly and daily input to aTS. PDFs are limited to the minimum and maximum $\Delta$KGE values and have been normalized with respect to its width to ease the comparison. A thin dashed horizontal line is given at $\Delta$KGE = 0 for reference. Dashed lines in the violins indicate the medians, dotted lines the $25^{th}$ and $75^{th}$ percentiles and solid lines the $5^{th}$ and $95^{th}$ percentiles.

The adaptive time step procedure of aTS allows the user to choose different input time steps. This might be the case if input runoff is provided as an aggregate over a specific period, for example as daily runoff. aTS aggregates or disaggregates any given temporal resolution to the internal time step constrained by a Courant number of 1 (equation 6). Similar performances are achieved with aTS using either daily or hourly inputs across all basins in the German and European dataset at every spatial resolution (Figure 4). This is achieved by aggregation and disaggregation to the internal time step but it is also affected by the fact that we compare against observed daily discharge. Sub-daily differences are thus averaged out before comparison. Observed hourly discharge would contain information about sub-daily variability that could not be obtained from daily inputs and, thus, hourly inputs might perform better in this case. However, observed discharge is mainly available on a daily resolution.

Evidently, aggregated input provides less variable runoff to the routing scheme, leading to less variable river discharge. Aggregation does, however, not change absolute values (bias). The $\Delta$KGE values therefore appear due to changes in streamflow variability, which should be reduced with aggregated runoff values.

Subtle differences exist between the $\Delta$KGE values for the German and European dataset. The median $\Delta$KGE values are almost zero for the European basins (Figure 4b) with very low standard deviations. Median $\Delta$KGE values for the German dataset are in contrast slightly negative around $-0.005$ (Figure 4a). The differences between the German and European dataset come mainly from the spatial resolution at which gridded runoff inputs for mRM were generated. Forcing for mRM was provided at 4 km resolution for the German dataset, which is the lowest resolution of the meteorological input (Zink et al.,

2017). The input runoff for mRM has been generated at a 24 km resolution for the European dataset, which corresponds to the resolution of the meteorological E-OBS dataset (Haylock et al., 2008). Runoff data at 4 km scale exhibit much higher spatial variability compared to the coarser 24 km runoff. The higher spatial variability of the German dataset is substantially reduced when using daily runoff compared to hourly runoff, which generates the small mismatch between using hourly and daily inputs for the German dataset (Figure 4a). The equalization of variability from averaging is less pronounced in the less variable runoff fields of the European dataset.

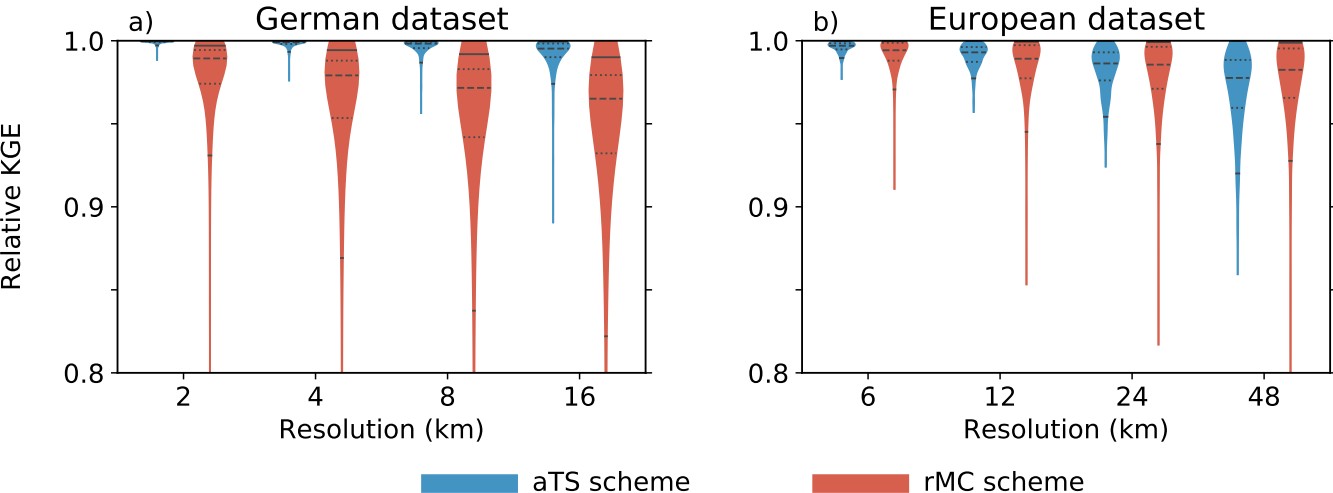

**Figure 5.** Probability density functions (PDF) of relative KGE values for adaptive time step scheme (aTS, blue violins) and the regionalized Muskingum-Cunge (rMC) routing scheme (red violins). KGE values are calculated relative to the finest possible spatial resolution, which is 1 km for the German (a) and 3 km for the European dataset (b). PDFs are limited to the minimum and maximum relative KGE values and are scaled to the same widths. Dashed lines in the violins indicate the medians, dotted lines the 25$^{th}$ and 75$^{th}$ percentiles and solid lines the 5$^{th}$ and 95$^{th}$ percentiles.

For the spatial scaling, KGE values relative to the finest possible model resolutions (1 km for German and 3 km for European dataset) are reported (Figure 5). In other words, the reference values (observations) in the calculations of KGE are replaced by simulated streamflow obtained with optimized parameters at the highest resolution. Perfect spatial scaling is hence indicated by a relative KGE value of one. Figure 5 shows the probability density functions of the relative KGE values estimated over all basins for each model resolution. The optimized parameter obtained at the highest spatial resolution for each basin is transferred for the aTS and the rMC parametrization to the model runs at the coarser spatial resolutions.

Results shown in Figure 5 clearly demonstrate a remarkable spatial scalability of aTS in comparison to the original rMC parametrization. The lowest median relative KGE of 0.977, which represents a change of less than 3 percent, is observed at the coarsest resolution of 48 km for the European dataset. The overall lowest relative KGE is 0.85 for aTS and 0.22 for the rMC scheme. The aTS scheme shows an improved scalability because it considers the between-grid heterogeniety of celerities through the parametrization based on terrain slope (equation 5) and the numerical stability criteria (equation 6). The

spatial scalability of aTS is higher for the German dataset compared to the European one. This can be attributed to the spatial resolution of the slope data used in the parametrization of celerity (equation 5), which is available at 100 m resolution in the German dataset compared to 500 m in the European dataset. The representation of river slopes is thus more realistic in the German dataset. Notably, a similar spatial scalability is found for both aTS and rMC parametrization if default parameters are used (not shown).

The adaptive time step scheme (aTS) shows, in summary, remarkable temporal and spatial scalability in comparison to its predecessor. The adaptive time step allows for aggregated or disaggregated input (generated runoff) from any given temporal resolution.

## 3.3    Comparison of Adaptive Time Step Routing with Regionalized Muskingum-Cunge Parametrization

The adaptive time step scheme (aTS) is the successor of the regionalized Muskingum-Cunge (rMC) routing implemented in mHM. A detailed analysis of the differences in model performances between the two routing parametrizations is presented here for the German and European dataset and selected spatial resolutions (Figure 6).

The performances are comparable across the German and European dataset (Figure 6), if aTS and rMC are calibrated individually on each basin and at each resolution. However, the cumulative distribution functions (cdfs) of $\Delta$KGE values are skewed towards positive values indicating in general higher performance for aTS than rMC (Figure 6a-6j, solid blue line in right panels). This improvement is slightly higher for the German dataset compared to the European, which can be attributed to the higher spatial resolution of the slope data in the former. $\Delta$KGE values are closer to zero for resolutions finer than 4 km indicating a more comparable model performance for aTS and rMC at higher spatial resolutions than at coarser ones. This is due to the fact that the original rMC routing scheme was developed at this resolution (Samaniego et al., 2010; Kumar et al., 2013b). At spatial resolution coarser than 12 km, the rMC routing strongly violates the Courant-Friedrichs-Lewy criterium (i.e., $c\Delta t/\Delta x \leq 1$) which results in poorer performance. Even re-optimizing the routing parameters cannot compensate for the scaling error because water is moved too fast through the river network. At these coarse resolutions, the aTS scheme is still outperforming the rMC scheme when run with a constant $\gamma = 15$ parameter for all catchments (Figure 6a-6j, dashed red line in right panels).

In summary, the adaptive time step scheme (aTS) demonstrates at least the same performances as its calibrated predecessor, the regionalized Muskingum-Cunge routing scheme (rMC). The scalability of mHM across spatial resolutions has been demonstrated before, but employing a fixed spatial routing resolution for the rMC scheme (see  Kumar et al., 2013a). For this purpose, the gridded runoff fields are spatially up- or down-scaled to the specified spatial resolution (e.g., 8 km runoff field disaggregated to 4 km). The aTS parametrization allows the user to simultaneously scale both the hydrologic and routing model. Notably, aTS requires no specific up/downscaling of runoff fields and parameters can be transferred across spatial and temporal resolutions. Both of these properties offer distinct advantages in reducing the computational costs because mRM can be directly applied at the resolution of the gridded runoff input. Using a constant $\gamma = 15$ parameter for all catchments, avoiding optimization, further reduces the computational costs but might result in slightly decreased model performances in comparison to model runs with optimized $\gamma$ values ($\Delta$KGE $\approx$ 0.1 at a 95% confidence interval). This is, however, still small compared

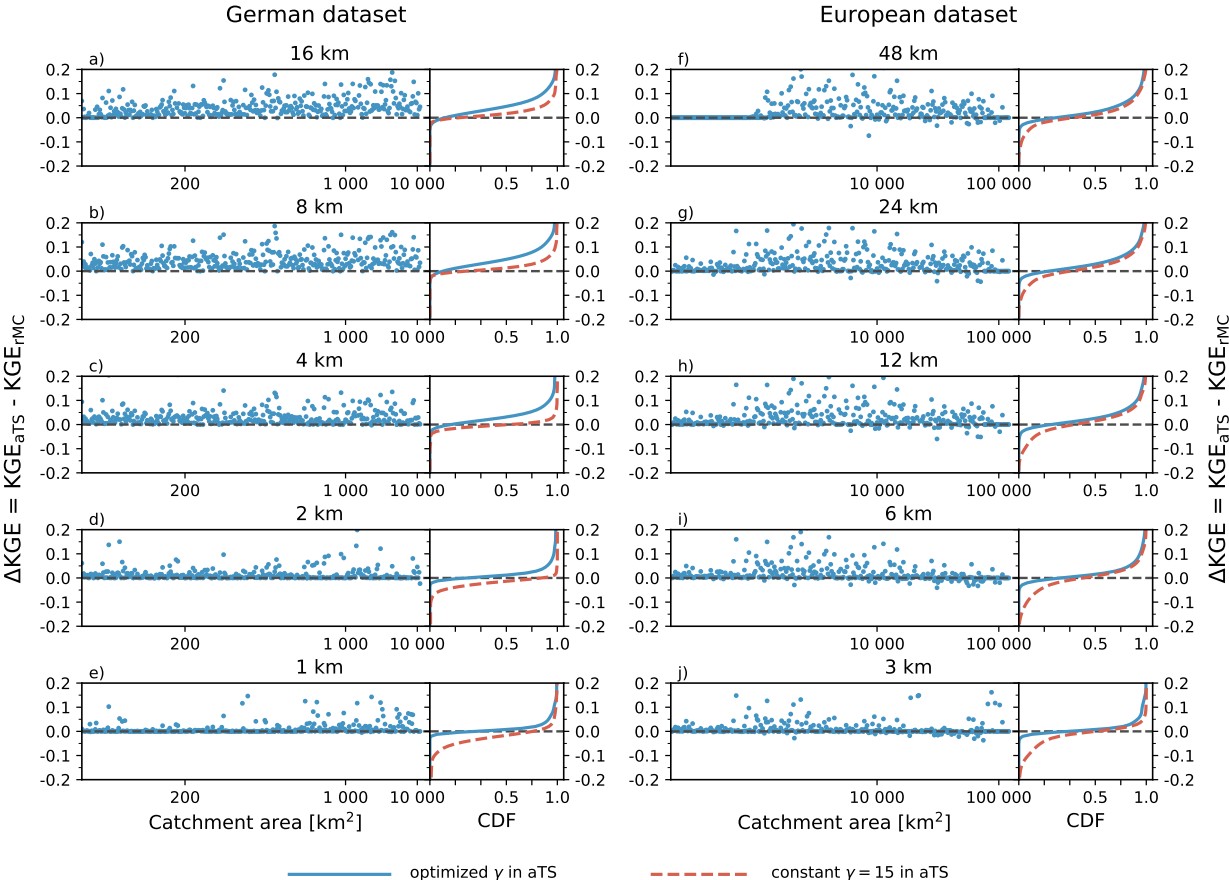

**Figure 6.** Differences in KGE between the multiscale Routing Model mRM using a optimized parameter $\gamma$, constant parameter value $\gamma = 15$ and the original Muskingum-Cunge routing scheme (rMC) implemented in the mesoscale Hydrologic Model mHM for the German (left column) and European dataset (right column). $\Delta$KGE values between aTS and rMC using optimized parameter values on each basin and at each resolution are shown for the respective basins on the left of each panel, where basins are sorted according to catchment area (note the logarithmic scale). Cumulative distribution functions of $\Delta$KGE between aTS and rMC using optimized parameter values (solid blue line) and aTS with constant parameter and rMC with optimized ones (dashed red line) are depicted on the right of each panel. The line at KGE = 0 (dashed black line) is added for reference.

to the impact that the hydrological model used as input to the routing scheme has. Using fixed $\gamma$ parameters also enables the seamless application of aTS at ungauged basins (Rakovec et al., 2016).

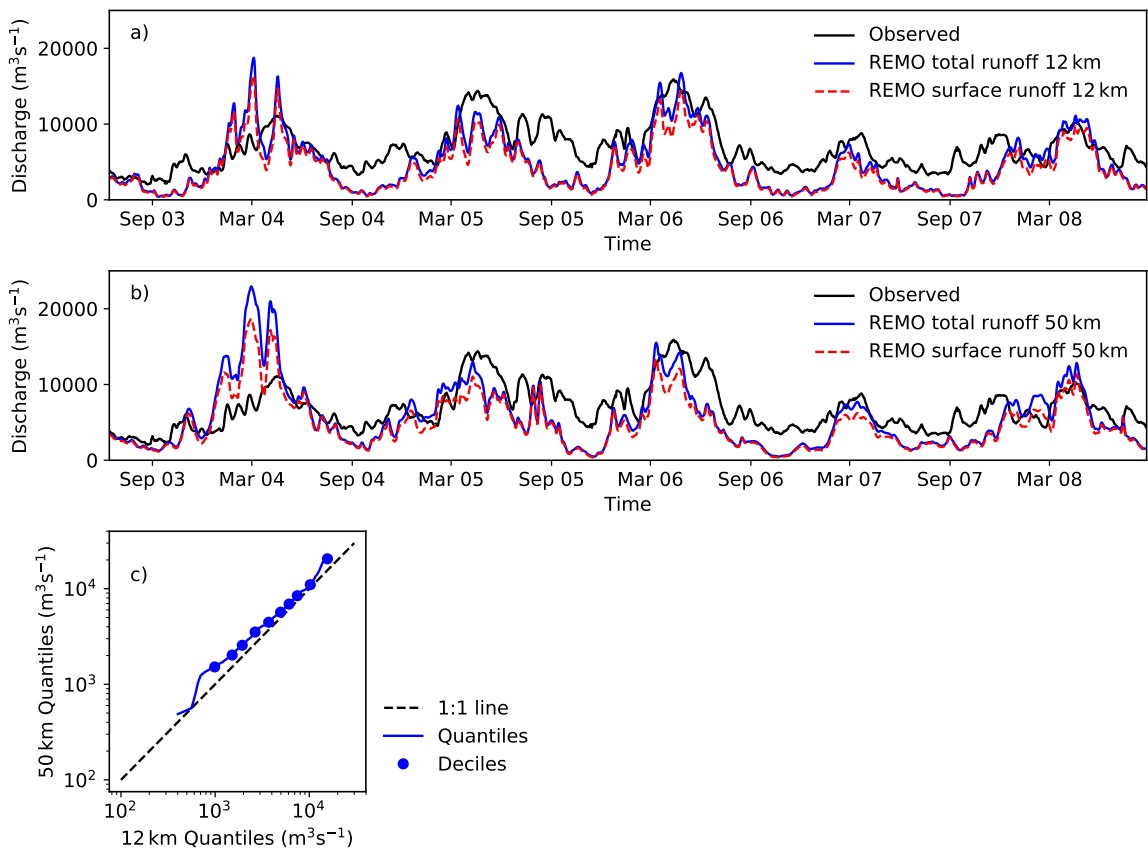

**Figure 7.** Hydrographs for the Danube river basin at the gauging station Ceteal-Izmail obtained by routing runoff output from the regional climate model REMO with mRM, employing the adaptive time step parameterization (aTS) at spatial resolutions of 12 km (a) and 50 km (b). The solid black line show the observed streamflow. The solid blue line is mRM output using as input the sum of the REMO surface and subsurface runoff components. The dashed red line is mRM output using only REMO's surface runoff component. Panel c) Q-Q plot of routed REMO outputs (sum of surface and sub-surface component) at 12 km and 50 km resolutions.

## 3.4 Streamflow Simulations over the Danube Catchment by applying mRM to the Regional Climate Model REMO

Regional Climate Models (RCMs) are used to dynamically downscale Global Climate Models over a specific region to obtain higher resolution information about the local climate. The evaluation of RCMs often focuses on surface fluxes and states, such as 2 m air temperature, precipitation, and evapotranspiration amongst others. River runoff, which provides an integrated signal of the water cycle over a region, is not often used as a further model diagnostic. This might be, besides other reasons, due to the fact that RCMs are designed to be run at various spatial resolutions, ranging from few kilometers (e.g., Jacob

et al., 2014) to tenth of kilometers (e.g., Van der Linden and Mitchell, 2009). RCM output has hence to be aggregated or disaggregated to current routing schemes with fixed routing resolutions. This is not necessary with the multiscale Routing Model mRM employing the adaptive time step parameterization (aTS) that runs seamlessly at various spatial and temporal scales (section 3.2). This eases the comparison of RCM-derived streamflow with observations as the routing model has to be

setup only once and can then be applied at different resolutions without adjusting the model parameter $\gamma$ or the model setup.

This section shows one exemplary application of mRM to output of the Regional Climate Model REMO (Jacob et al., 2001) over the Danube catchment. Generated runoff by REMO has been obtained from the EURO-CORDEX project (Jacob et al., 2014) at 12 km and 50 km resolutions. Both resolutions were used to run mRM employing the aTS parameterization ($\gamma = 15$) over the Danube catchment (Figure 7a for 12 km and 7b for 50 km). REMO was nested into ERA-Interim reanalysis in the

EURO-CORDEX project, which therefore permits the comparison against observed streamflow. The Danube basin is part of the European dataset used for the evaluation in previous sections. The same setup was used for routing gridded runoff fields of the REMO model and only the number indicating the routing resolution had to be changed in the mRM configuration file.

There are striking differences between the observed and simulated streamflow (Figure 7). This is due to the fact that REMO uses a simple runoff generation consisting of direct surface runoff and soil drainage. There is no groundwater description in

REMO so that the only water storage is in the river itself and baseflow is much too low. This becomes evident as most runoff generated by REMO is surface runoff. REMO at 50 km resolution shows substantial contribution from subsurface runoff only during spring 2004 (Figure 7b). This highlights a common misunderstanding when using river routing with land surface models: most land surface models include only a simple runoff generation that does not account for the temporal variability of the runoff signal, which is separated into fast flow, interflow and baseflow in hydrologic models (e.g., mHM). Routing

directly drainage fluxes directly leads to seasonal high flows that are much earlier than observed. Using very low celerities in the routing models might improve this model-data mismatch (see Oki et al., 1999, for celerities in several routing schemes). Other runoff schemes represent different runoff components within the model code (e.g., Lohmann et al., 1996; Hagemann and Dümenil, 1997; Pappenberger et al., 2009). The multiscale Routing Model mRM does not contain runoff generation because most hydrologic models already include detailed runoff generation and also land surface models start to include groundwater

components nowadays (e.g., Niu et al., 2011; Clark et al., 2015). Details of these components depend strongly on model focus which should not be imposed by the river routing model (cf. section 4).

Three main conclusions can be drawn from the comparison of modeled and observed discharge: first, REMO is able to capture the overall seasonal variations of runoff. There is a pronounced seasonality within the first three years in both, observation and REMO simulated streamflow, which is much reduced in the last two years (Figure 7a & b). Seasonality in the

Danube catchment is dominated by spring melt, which is very low in the last two years. REMO is therefore able to simulate inter-annual variations in precipitation and surface temperature over the Danube catchment.

Second, REMO produces too little runoff on average at both resolutions. Runoff is underestimated by about 30% and 17% on 12 km and 50 km resolutions, respectively. Interestingly, REMO overestimates catchment average precipitation compared to E-OBS (Haylock et al., 2008) by 2% and 15% on 12 km and 50 km, respectively. Hence, the partitioning of precipitation

into runoff and evapotranspiration is not correct in REMO, under the reasonable assumption that groundwater tables around

the Danube river exhibit no significant trend over the simulation period. This implies that evapotranspiration is overestimated very similarly on 12 km and on 50 km resolution by REMO.

Third, REMO exhibits statistically different runoff on both model resolutions (i.e., 12 km and 50 km). The quantile-quantile plot (Figure 7c) shows that the 50 km simulation produces more runoff than the 12 km simulation, most likely because REMO simulates also higher precipitation on 12 km resolution. It is worth noting that this mismatch is not present in the ENSEMBLES project (see Appendix D).

This section underlines the fact that hydrologic and land surface models have to include the processes of runoff generation and groundwater for a fair comparison of modeled and observed streamflow. Process parametrizations with an instantaneous surface runoff component are common in land-surface models (Vereecken et al., 2019), but they are too inflexible to reproduce observed discharge. After these process parametrizations account for more runoff components (e.g., fast and slow interflow), the multiscale Routing Model mRM would allow to further analyse the responses of land surface models to climatic extremes (Reichstein et al., 2013) using indices and signatures of the discharge time series (Thober and Samaniego, 2014; Shafii and Tolson, 2015).

## 4    Comparison with Existing Routing Schemes

River routing is performed at various resolutions depending on the application. Global streamflow simulations, using output of land-surface models (LSMs) or hydrological models (HMs) for example, are typically carried out at 0.5° or 1.0° resolutions (a.o., Oki et al., 1999; Hagemann et al., 2009; Pappenberger et al., 2009; Zhang et al., 2016). However, climate models are run on ever increasing spatial scales (Jacob et al., 2014), or using even internally nested grids or zooming functionality (Zängl et al., 2014). Also, spatial resolutions of few kilometers are used within the hydrologic modeling community to obtain national and continental estimates of hydrologic fluxes and states (David et al., 2011; Marx et al., 2018; Thober et al., 2018; Wanders et al., 2019). Despite the fact that diverse spatial resolutions are used to represent the hydrologic cycle, spatial resolutions of routing are mostly fixed and cannot be changed easily. In many models, the user needs to provide the input data (e.g., flow direction, DEM, channel information) for every resolution the model is applied on (a. o., Lohmann et al., 1998; Beighley et al., 2011; Neal et al., 2012). The multiscale Routing Model mRM, on the other hand, is able to scale the river network to the desired routing resolution internally. This allows to make full use of the information provided by the input runoff data, without uncertainties coming from the rescaling process (e.g., from a 12 km LSM output to a 0.5° river network). It also avoids scaling the input runoff to a hyper-resolution river network, which then requires high-performance computing resources such as in the case of the RAPID framework (David et al., 2011). This might especially be valuable if parameter estimations using discharge data need to be carried out, which requires a large amount of model evaluations.

Current solvers describing water movement within a river network can in principle be applied at different resolutions. For example, the solution of the diffusion equation by Greens functions proposed by Lohmann et al. (1996) is valid independently of the resolution of the river network. The CaMa-Flood model can similarly be applied to different resolutions as long as the Courant-Friedrichs-Lewy condition is fulfilled (Yamazaki et al., 2013). aTS employs the same condition to identify an adaptive

time step that guarantees the numerical stability and achieves a scalability across spatial resolution. Yamazaki et al. (2009) also developed a pre-processor for the CaMa-Flood model that explicitly allows to generate a river network at different spatial resolutions. mRM follows the same idea but it internally includes the upscaling of the river network to the required resolution in the model code. The user has to provide the routing network only once even if the application will focus on different spatial resolutions. The derived river network can be stored in a restart file to further speed-up the computation (see Appendix B for run times). However, aTS performance is dependent on the resolution of the underlying slope data (see Section 3.1 and 3.2) and it is advisable to use the highest resolution data available. This is due to the fact that channel slope instead of terrain slope should be used in equation (5) and a high-resolution DEM provides a close approximation of channel slope.

Another reason that hampers the scalability of existing routing models is that they include not only the routing of water in the river network but also a runoff generation mechanism, which represents a variety of other components of the hydrologic cycle (Pappenberger et al., 2009). The complexity of existing runoff generation descriptions reflects the diversity of use cases of hydrologic and land surface models. Descriptions range from simple linear models (Niu et al., 2011; Beven, 2012) to more complex representations considering surface groundwater interactions (Maxwell and Kollet, 2008; Miguez Macho and Fan, 2012). Existing routing schemes often opt for more simple parsimonious representations. For example, routing models use linear reservoirs for overland flow, baseflow and riverflow to delay runoff generated by the land surface (e.g., Hagemann and Dümenil, 1997; Pappenberger et al., 2009; Getirana et al., 2012). mRM does not include any runoff generation because it is beyond the scope of a river routing model to reflect the complexity of existing runoff generation processes. Notably, there is currently ongoing research in understanding how a particular process parametrization impacts hydrologic simulations (Niu et al., 2011; Clark et al., 2015). Runoff generation also hampers the scalability of routing models because of their highly non-linear behavior. The Multiscale Parameter Regionalization (MPR, Samaniego et al., 2010) is one of few approaches that has proven to provide consistent generated runoff at resolutions ranging from 2 km to 16 km for mesoscale catchments (Kumar et al., 2013b) and from $0.125°$ to $1°$ for continental scale basins (Kumar et al., 2013a).

Among the plethora of routing models presented over the past decades, only few have rigorously evaluated their spatial scalability. The "Model for Scale Adaptive River Transport" (MOSART) has been developed explicitly to achieve seamless application of river routing across scales (Li et al., 2013), similar to mRM. MOSART has been successfully coupled, for example, to the Community Land Model to compare with global discharge data (Li et al., 2015). MOSART differs from mRM by solving the kinematic wave equation with Manning's equation for channel velocity (Manning, 1891) not only for the main channel but also for hillslope routing and subgrid tributaries. It thus explicitly accounts for sub-grid heterogeneity by considering all lateral travel times across hillslopes and tributaries. mRM, on the other hand, solves a kinematic wave equation with spatially varying velocities for the main channel only (equation 1 and 5). The assumption within mRM is that travel times in the main channel dominate travel times at hill slopes and tributaries and the latter are negligible. This, in turn, leads to a simpler approach with one model parameter. However, further research is needed to explicitly investigate the validity of this model assumption. It is for example possible to return to the original formulation of Miller et al. (1994), using a reference slope $s_0$, that should depend on the underlying digital elevation model (DEM). But two DEM resolutions, as in this study, are not

enough to find a suitable formulation for the dependence of $s_0$ on DEM characteristics such as resolution or maximal slope. $s_0$ was hence lumped with the minimum celerity $c_0$ to give only one identifiable parameter $\gamma$.

It is worth reminding that mRM represents a simple approach towards river routing. The results in this study demonstrate that mRM employing the adaptive time step parameterization in combination with upscaled high resolution celerities (aTS) achieves almost identical daily streamflow simulations at various model resolutions in diverse German and European catchments. Recent literature has shown that a realistic representation of streamflow in river basins with extensive floodplains such as the Amazon, Niger, and Congo river require the representation of floodplain inundation processes (Getirana et al., 2012; Paiva et al., 2013; Fleischmann et al., 2016; Pontes et al., 2017). Floodplain processes are currently not considered in mRM and further research is required to include these. Paiva et al. (2013) showed that floodplain processes dominate the difference between a hydrodynamic and kinematic wave models. The approach used therein should be exploited within mRM to be applicable at different resolutions.

## 5 Conclusions

The adaptive time step scheme in combination with upscaled high resolution celerities (aTS) implemented in the multiscale Routing Model mRM estimates streamflow at various resolutions ranging from the hyper-resolution of 1 km to the large scale of 0.5°. Differences in Kling-Gupta efficiencies of simulated daily streamflow between various model resolutions and temporal forcings are negligible with a median of 0.03 over Germany and Europe (Section 3.2). The aTS scheme shows an improved scalability over its predecessor because it considers the linkage between spatial resolution and integration timestep by virtue of the Courant criteria (equation 6). It considers the between-grid heterogeniety of celerities through the parametrization based on high-resolution terrain slope (equation 5). mRM represents the river network internally at the resolution of the model input, which allows seamless application to output of any hydrologic model (HM) and land surface model (LSM). It can also easily be coupled internally in the code of HMs or LSMs, providing error measures such as Nash-Sutcliffe and Kling-Gupta efficiencies for model evaluation or calibration.

mRM uses a simple kinematic wave equation to describe water flow within a river network. This representation is regarded suitable as long as backwater effects and floodplain inundation processes are comparatively small. mRM does not represent runoff generation mechanisms, which are included in other routing models. Runoff generation is included in hydrologic models and nowadays often in land surface models. The details of the implementation depend strongly on the application of interest. Users of river routing schemes should not be limited by the options implemented in the river routing model itself.

mRM can in principle be used on rotated model grids commonly used for climate models if high resolution flow directions are provided at the same grid. However, mRM represents the river network as a rectangular grid, allowing to apply a constant time step over the entire model domain. Future developments will focus on implementing reservoirs and natural lakes, floodplain processes, and a location dependent time stepping scheme. The latter will enable the use of mRM on irregular grids or in models with local refinement. Also, parallelisation is currently implemented in mRM to take full advantage of high-performance

computing clusters. The model source code along with a test case to validate succesful installation is freely available within the codebase of the mesoscale Hydrologic Model mHM at www.ufz.de/mhm.

*Code availability.* The software code is available through a public git repository hosted at the Helmholtz-Centre for Environmental Research - UFZ with the url https://git.ufz.de/mhm/mrm/. The software version used for this paper can also be identified by the git tag "mRMv1.0".

The manual of mHM contains a chapter on the installation and user guide of mRM (Chapter 9) and the full mHM manual is also contained in the mRM git repository. Input and output data of mRM is also included in the git repository to test succesful installation (see manual on how to run the test basin).

## Appendix A:  Regionalized Muskingum-Cunge (rMC) routing

The regionalized Muskingum-Cunge (rMC) parametrization implemented in the mesoscale Hydrologic Model mHM calculates

the Muskingum coefficients $C_1$, $C_2$, and $C_3$ in equation (3) as a function of high-resolution river network properties. The coefficients $C_1$, $C_2$, and $C_3$ are parametrized as follows

$$C_1 = \nu_2; \ C_2 = \nu_1 - \nu_2; \ C_3 = 1 - \nu_1, \tag{A1}$$

where the parameters $\nu_1$ and $\nu_2$ are given as

$$\nu_1 \quad = \frac{\Delta t}{\beta(1-\epsilon)+\frac{\Delta t}{2}}; \tag{A2}$$

$$\nu_2 \quad = \frac{\frac{\Delta t}{2}-\beta\epsilon}{\beta(1-\epsilon)+\frac{\Delta t}{2}}$$

following the nomenclature of appendix A2 in Samaniego et al. (2010). This formulation is identical to equation (4) of the present study, using $\beta = \Delta x / c$ in equation (A2) and substituting equation (A2) into equation (A1). The parameters $\beta$ and $\epsilon$ are then conceptualized as

$$\beta \quad = \gamma_1 + \gamma_2 L + \gamma_3 S + \gamma_4 C; \tag{A3}$$

$$\epsilon \quad\quad = \gamma_5 \frac{S}{max(S)},$$

where $L$ is the length of the reach, $S$ is the slope of the reach, and $C$ is the fraction of impervious land cover within the floodplains (see table 4 in Kumar et al. (2013b)). Overall, there are five global parameters $\gamma_1$ to $\gamma_5$ in equation (A3) that can be chosen by the user. The integration time step is fixed at one hour. To guarantee the numerical stability of the parameterization, the following upper and lower bounds are applied

$$0 \quad < \epsilon \leq 0.5, \tag{A4}$$

$$\frac{\Delta t}{2(1-\epsilon)} \quad < \beta \leq \frac{\Delta t}{2\epsilon}, \tag{A5}$$

where $\Delta t$ is set to one hour.

## Appendix B: Run times

The run times of mRM do not scale linearly with the number of grid cells. The reason is that the arrays containing the network information cannot be stored continuously in memory because the river network can be mathematically represented as a tree. The run times for a small and large basin are reported here to provide an overview of the range of possible run times. The
Moselle catchment with an area of 28 286 km$^2$ is selected to represent a small catchment. A spatial resolution of 24 km results in 34 grid cells to cover the Moselle catchment. The Danube river with an area of 801 463 km$^2$ is selected to represent a large catchment. The REMO simulations (Section 3.4) at 12 km resolution resulted in 5775 grid cells.

The run time has to be separated for the initialization and computation step of mRM. During the initialization step of mRM, all input data is read and the high resolution river network is upscaled to the model resolution specified by the user. mRM
offers restart capabilities that allows the user to perform this step only once. The initialization of mRM takes about 1.3 s and 3300 s for the Moselle and Danube river, respectively. It heavily depends on the speed of I/O because all the data is read during this step and the cache size of the employed CPU. If mRM reads the upscaled river network from a restart file, this step takes a negligible amount of time. For example, the initialization step takes 60 s for the Danube river if a restart file is used. During the computation step of mRM, the streamflow values within the river network are calculated. The run time of this step
scales linearly with the length of the simulation period. This step takes 0.1 s and 24 s per year for the Moselle and Danube river, respectively. These run times have been estimated with the Intel 18 fortran compiler and level 3 code optimization on a Dual Intel Xeon Platinum 8169 CPU (http://www.fz-juelich.de/ias/jsc/EN/Expertise/Supercomputers/JUWELS/Configuration/ Configuration_node.html).

## Appendix C: Drainage area for different model resolutions

The D8 method (O'Calaghan and Mark, 1984) is known to be unable to reproduce basin area correctly at large scales (e.g., 1°). This effect can also be seen for mRM in four major European river basins (Figure C1). The setup for this analysis is the same as the one described in Samaniego et al. (2018, see data availability section). Two use cases of mRM are investigated here: first, a single-basin setup where the entire model domain drains to one outlet; second, a continental-scale setup that contains multiple rivers and the model domain contains multiple outlets. In the former case, the basin area calculated in mRM is independent of
the model resolution and equal to the basin area at the high-resolution input data (0.5 km in this case). This is achieved by using weighted area fractions for grid cells that are only partly covered by the study domain. The calculated basin area is very close to the true basin area and differences stem from mismatches in the delineation of the basin. In the second case, the basin area is correctly reproduced up to a model resolution of 40 km and tends to increase for lower model resolutions (blue markeres in Figure C1). This effect is larger for small basins (e.g., the Elbe river and Loire river) than for large basins (e.g., the Danube
river) because it enlarges as the ratio between basin size and model resolution decreases. The increase of basin area with the size of grid cells can be expected because larger grid cells have longer edges and thus unify more head water streams. If the underlying rivers do not unite, but depart downstream, then water is not routed correctly and no meaningful analysis can be carried out. Yamazaki et al. (2009) provide an insightful illustration of this deficiency of the D8 method the 1° resolution. They

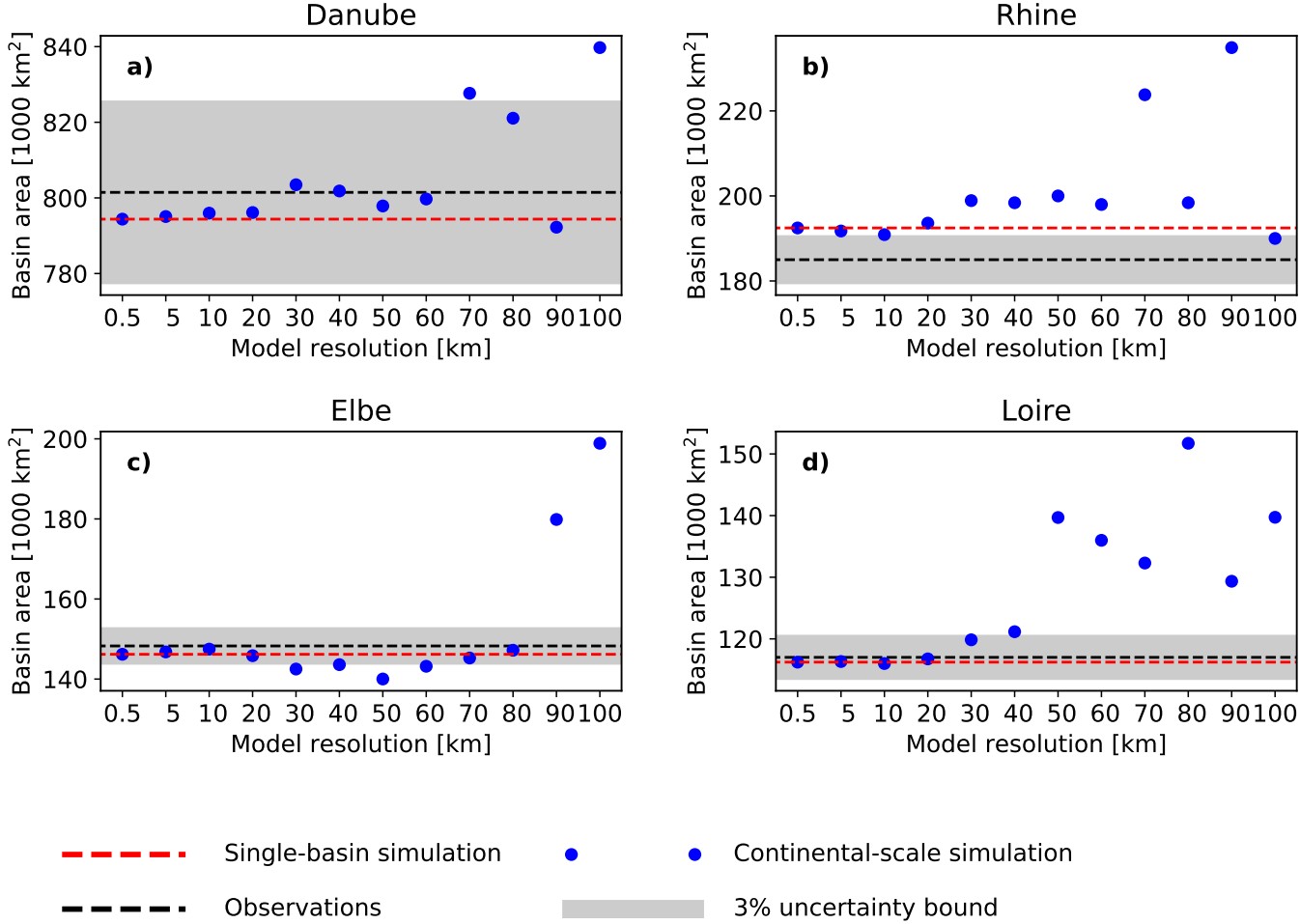

**Figure C1.** Basin areas for four major European river basins. Blue circles denote the calculated basin area derived by the D8 method at different resolutions for continental scale simulation. Red dashed line shows the basin area for a single-basin setup. Black dashed line show the true value with an uncertainty bound of 3% around it.

also proposed an improvement of the D8 method to correctly route water at this coarse scale and provide an example for the Mekong, Salween, and Yangtze rivers (see Figure 6 in Yamazaki et al., 2009). This improvement is not implemented in mRM because firstly, basin area is correctly represented in single-site setups, which are frequently used for parameter estimation. Secondly, mRM can be run at high resolution of less than 40 km even if the input is provided at coarser scale (e.g., 100 km). mRM distributes the input equally among all high-resolution grid cells and then routes the water downstream. mRM is also computationally efficient to simulate streamflow at high resolution of 5 km over continental scales in the context of climate change studies (e.g., Thober et al., 2018; Marx et al., 2018) and seasonal forecasting (Wanders et al., 2019). Notably, the calculated flow accumulation of mRM is saved in a restart file. The user can thus easily check whether the chosen model resolution adequately represents the actual river basin size (i.e., within the acceptable error bounds).

## Appendix D: REMO simulations from the ENSEMBLES project

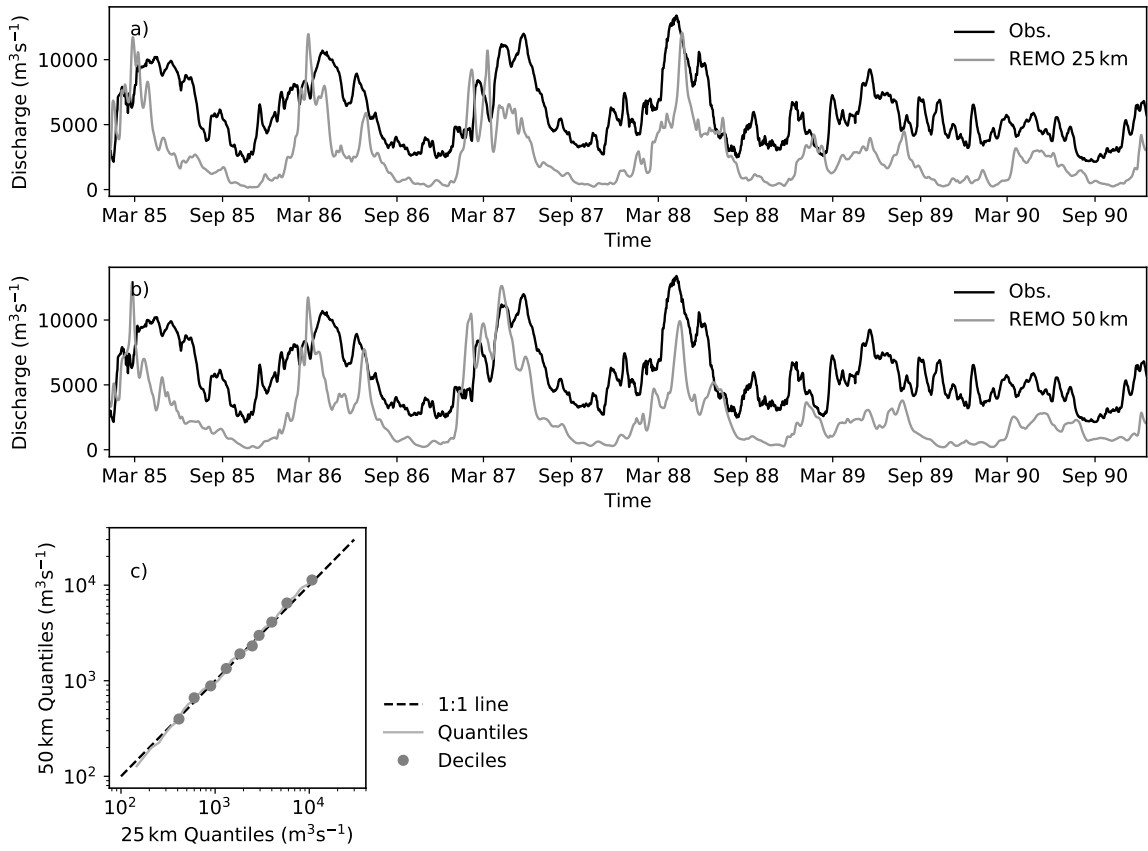

**Figure D1.** Same as Figure 7, but using REMO simulations from the EURO-CORDEX project.

The same simulations as in Section 3.4 have been conducted with the REMO simulations created within the ENSEMBLES project. There are two main differences between the ENSEMBLES and the EURO-CORDEX simulations. First, the higher resolution is 25 km in ENSEMBLES whereas it is 12 km in EURO-CORDEX. Second, ERA-40 is used as boundary condition in the ENSEMBLES project whereas ERA-INTERIM is used in EURO-CORDEX. Interestingly, the bias of precipitation is less than 3% for both resolutions used in the ENSEMBLES project whereas it is 15% for the 50 km run of EURO-CORDEX. However, the ENSEMBLES simulations show an underestimation of observed streamflow by up to 50% (Figure D1). This bias is reduced in the EURO-CORDEX simulations. A commonality between the EURO-CORDEX and ENSEMBLES simulations is that the absolute bias in streamflow is larger than the absolute bias in precipitation. This highlights that REMO overestimates evapotranspiration.

*Author contributions.* ST, MC, and LS designed the study. ST, MK, and JM conducted parameter estimation and model validation. RK and LS provided insights into rMC parametrization. All authors contributed to the writing of the manuscript.

*Acknowledgements.* This study has been partially funded within the scope of the HOKLIM project (www.ufz.de/hoklim) by the German Ministry for Education and Research (grant number 01LS1611A). This study has been partially funded by the Copernicus Climate Change

5   Service. ECMWF implements this Service and the Copernicus Atmosphere Monitoring Service on behalf of the European Commission. The study is a contribution to the Helmholtz Association climate initiative REKLIM (www.reklim.de). MC was supported by a grant overseen by the French National Research Agency (ANR) as part of the "Investissements d'Avenir" program (ANR-11-LABX-0002-01, Lab of Excellence ARBRE). The ENSEMBLES data used in this work was funded by the EU FP6 Integrated Project ENSEMBLES (Contract number 505539) whose support is gratefully acknowledged. We acknowledge the E-OBS dataset from the EU-FP6 project ENSEMBLES

10  (http://ensembles-eu.metoffice.com) and the data providers in the ECA&D project (http://www.ecad.eu). We also acknowledge our data providers: the European Environment Agency, the Harmonized World Soil Database, the Global Runoff Data Centre, German Meteorological Service (DWD), the Joint Research Center of the European Commission, the Federal Institute for Geosciences and Natural Resources (BGR), the Federal Agency for Cartography and Geodesesy (BKG), and the European Water Archive. The data used within the European dataset are described in Rakovec et al. (2016) and the data used within the German dataset are described in Zink et al. (2017). Further simulation data

15  that supports findings of this study are available from the corresponding author upon request. We thank the editor Jeffrey Neal for handling our manuscript and Thomas Riddick and one anonymous reviewer for their constructive comments that helped to substantially improve our model and manuscript.

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
