# Peer review of "The multiscale Routing Model mRM v1.0: simple river routing at resolutions from 1 to 50 km"

_Geoscientific Model Development, 2019_

## Referee Comment (RC1) · Anonymous Referee #1 · 13 Feb 2019

**1   General remarks**

The study presents the mRM which specialized in the routing of main channel flow. It is using an adaptive time step and scales well for different resolutions. It is able to upscale river flow networks and parameters to the input data resolution by itself and thus provides a very user friendly tool for the computation of river discharge from different models.

The manuscript is very well written and structured. The reported analysis convincingly support the conclusions. Limitations (e.g. missing floodplain processes) are listed

together with future plans. For LSMs without a native routing scheme, the mRM is definitely an interesting tool although the apparent limitation of using only total runoff instead of the different components might limit its applicability somewhat. It would be nice if the authors could add a short paragraph about the resolution limit on the coarse side. Large scale earth system models are usually much coarser than 50km and focus on routing the discharge into the correct ocean basin rather than using it for evaluation. Would this tool be applicable at such resolutions as well?

**2 Specific remarks**

- P2L12: I don't understand this sentence. I guess you mean you provide a framework for those LSMs without a native river routing scheme to compute river discharge and compare it to observations? Please rephrase.

- P17L1: Would this mean that using mRM with a LSM that generates multiple runoff components, e.g. fast runoff, baseflow..., that mRM would have to be applied separately for each of them with a component specific celerity? Please clarify.

- P17L1: REMO does separate runoff into two components: surface runoff and drainage. How are they treated for this study? Are they just combined to total runoff?

- P17L25: The statement seems a bit unfair. As just said REMO does compute different runoff components and (according to the output variable list) they should be available from the ENSEMBLES database. Also, ENSEMBLES is not exactly the newest project out there. Why not using data from the EURO-CORDEX Project which would allow to draw conclusions about REMO that would be much more up to date. ESGF has the variables total runoff (mrro) and surface runoff (mrros)

available (drainage (mross) would just be mrro – mrros for this model). Having said this, I like to stress that this does not compromise the manuscript in any way as the REMO analysis is just an example for the functionality of tmRM. Thus, no changes are necessary here.

- P21L26: Is this a left-over from an earlier submission? A bit early to thank for constructive comments before knowing what you get ;) . Still, I appreciate the attitude. Btw, at least I was contacted by a different editor and also I cannot see Paul Dirmeyer in the Editorial Board of GMD.

**3  Technical remarks**

- P1L12: are they really identical? I guess you mean similar, right?

- P1L14: everything is basically comparable. Do you mean similar again?

---

## Referee Comment (RC2) · Thomas Riddick (Referee) · 11 Mar 2019

**1 General comments**

This paper presents a model for simulating river discharge across a range of resolutions. This will likely be a very useful tool for a range of applications and its high degree of flexibility is a particular strong point. I generally find the model to be well presented and the validation provided on two different datasets to be thorough. The quality of the figures given is good and the quality of the writing generally high though there a few notable lapses (see technical corrections for those I have noticed).

My only concern is a lack of discussion of the upscaling of river directions themselves; particularly to the coarser scales. Upscaling D8 river directions can produce errors in routing that leads to water entering the wrong catchment. Did the authors encounter such problems? Would they expect to encounter them for river systems outside Europe? Were other upscaling algorithms (e.g. Yamazaki et al. 2009 "Deriving a global river network map and its sub-grid topographic characteristics from a fine-resolution flow direction map") considered at any point? Even if uncommon such routing errors would have a major impact on model performance so are worthy of some consideration/discussion.

2 Specific comments

Page 3 Line 2 – Minimization of computational demands is mentioned here but unless I missed it no indication of the actual computational demand of this scheme is mentioned here. Ballpark figures for a one or two resolutions and grid sizes would be useful for comparison purposes – how long does a simulation of X timestep/days/years take on machine Y? Is parallelisation possible?

Page 4 Line 7 – What does "center grid" mean here? Is this supposed to mean "grid center"?

Page 9 Line 19 – "exhibits limited impact to change of epsilon and gamma." – I am not quite sure what is meant here. Please review this sentence

Page 15 Line 31 – This sentence isn't clear. I assume it is meant that the model can also handle rotated grid as well as regular latitude longitude grids. Please clarify. Also, how about other grid e.g. a triangular grid?

Page 18 Line 7 – "It also avoids further computational demand by scaling the generated runoff etc. etc.". I wasn't clear what was meant by this sentence. Please clarify.

3 Technical corrections

Page 5 Line 9 – "the equation 5" -> "equation 5"

Page 5 Line 9 – Incomplete sentence "the sufficiently high model performance" -> "the sufficiently high model performance that it gives."

Page 5 Line 11 – I suggest "but it is not as available as…" -> "but it is not as readily available as…"

Page 6 Line 6 – "dividers" -> "divisions"

Page 7 Line 13 – "along -> "on"

Page 8 Line 6 – "daily resolution NetCDF files" -> "daily resolution in NetCDF files"

Page 9 Line 9 – "which was obtained" -> "which were obtained"?

Page 9 Line 13 – Invalid grammatical construction "because they allow to investigate" -> I suggest to replace it with "because they allow us to investigate"

Page 11 Line 23 – "aTS allows to use different" -> "aTS allows us to use different"

Page 12 Line 2 – "very little standard deviations" -> "very low standard deviations"

Page 12 Line 9 – "which generates the little mismatch" -> "which generates the small mismatch"

Page 13 Figure 5 Caption – "light gray violins", "dark gray violins". I see only red and blue violins. It seems the color scheme has been updated without updating the caption. Please update this caption.

Page 13 Figure 5 Caption – "and have scaled to the same widths" -> "and are scaled to the same widths"

Page 15 Line 12 – "The aTS parameterization allows to simultaneously" -> "The aTS parameterization allows us to simultaneously"

Page 15 Line 18 – "still little" -> "still small"

Page 15 Line 18 – "that has the hydrological model used as input to the routing

scheme." -> "that the hydrological model used as input to the routing scheme has."

Page 17 Line 12 – "very little" -> "very low"
* * *

---

## Short Comment (SC1) · 25 Mar 2019

Dear Reviewer #1,

We would like to thank you for your time and effort reviewing our manuscript. We see no major obstacle addressing your remarks in an improved version of our manuscript and would like to shortly outline the two main improvements we will implement in the following:

1.) Regarding the resolution limit of mRM: we have not tested beyond 50 km but are confident that it would perform well at 100 km resolution. However, it is important to

apply the model at a resolution where the river network correctly resolves the basin area at the mouth of the river. For example, this might not be the case using the D8 method implemented in mRM at 1 degree resolution (Yamazaki et al. 2009). And this might lead to errors in a large scale model, which could discharge streamflow into the wrong ocean basin. However, using mRM, the resolution at which the model is applied does not necessarily has to be the resolution of the input data. mRM can be forced with runoff at 1 degree resolution but perform the routing at 0.25 degree. It would internally scale the input to the model resolution. We will include a paragraph in the revised version to discuss this point.

2.) Regarding the use of ENSEMBLES data: We have chosen ENSEMBLES because it provided RCM output data at two resolutions, we had downloaded the data already, and we had used it in earlier publications. We are currently investigating the effort of using EURO-CORDEX. If REMO produces, however, similar results with EURO-CORDEX compared to ENSEMBLES, we will only add a note in the text and leave the section as it is. After all, section 3.4 is only a showcase of applications that can be easily conducted with mRM because the model only needs to be setup once regardless of the resolutions it is applied to.

References:

D Yamazaki, T Oki, and S Kanae. "Deriving a global river network map and its sub-grid topographic characteristics from a fine-resolution flow direction map." Hydrol Earth Syst Sci 13(11), 2241-2251, 2009. http://www.hydrol-earth-syst-sci.net/13/2241/2009/

---

## Short Comment (SC2) · 25 Mar 2019

Dear Thomas,

thank you very much for your detailed and insightful review. We see no problem improving our manuscript according to your comments and would like to shortly outline in the following how we intend to do so:

1.) Regarding upscaling using the D8 method: The correct representation of the drainage area at evaluation gauges is crucial for model validation. We did not encounter a large mismatch (>10\%) at the resolution that we investigated in this

manuscript. However, we would expect these errors to occur in the same way as they did in the work of Yamazaki et al. (2009). In particular in the region east of the Himalaya where three large rivers, the Mekong, the Yangtze, and the Salween, flow almost parrallel to each other over a few hundred kilometers, being less than 1 degree apart from each other (see Fig. 6 in Yamazaki et al. 2009). A mistake in the upscaling of the river network, particularly in this region, will be associated with large errors in any simulation. However, there is no need to run mRM at these coarse resolution of 1 degree. Even if the input to the model is provided at this coarse resolution, the model can be applied at a higher resolution of 0.25 degree where basin area is correctly reproduced. Run times also permit these applications (see comment below). We will put more emphasis on this point in the revised version of our manuscript and also highlight that the flow accumulation area of the upscaled river network is provided as output of the model.

2.) Regarding runtime and parallelization: We are currently parallelizing the routing which is the bottleneck in massive parallel hydrologic modelling at the land-surface, if groundwater is neglected. The parallelization of routing algorithms has to account for the river network because data between grid cells is only exchanged along the river. It is a non-trivial task to minimize communication and prevent idling of individual threads. The parallelization of mRM will be the subject of an upcoming study. It takes about 90 secondes to route 230 000 grid cells for 10 000 timesteps, which corresponds to 13 months simulation with hourly time steps, on a Dual Intel Xeon Platinum 8169 (http://www.fz-juelich.de/ias/jsc/EN/Expertise/Supercomputers/JUWELS/Configuration/Configuration_node.html) compiled with Intel Fortran and O3 optimization without any parallelization. These number does not account for the initialization of the model, which takes a substantial amount of time for 230 000 grid cells (i.e., continental-scale application), in particular the upscaling of the river network. This step, however, only has to be done once because the upscaled river network is saved in a restart file and can be read from it in subsequent applications. We will provide information on these aspects in a revised

version of our manuscript.

References:

D Yamazaki, T Oki, and S Kanae. "Deriving a global river network map and its sub-grid topographic characteristics from a fine-resolution flow direction map." Hydrol Earth Syst Sci 13(11), 2241-2251, 2009. http://www.hydrol-earth-syst-sci.net/13/2241/2009/

---

## Author Comment (AC1) · 6 May 2019

We thank the reviewers for their detailed comments, which helped us to further improve our manuscript. We attach a revised version of our manuscript with highlighted changes to this report as supplementary pdf. All line numbers below refer to this pdf with highlighted changes. Among the changes we made, the following are the most important ones:

- We updated the REMO run used in Section 3.4 to the simulations carried out in the EURO-CORDEX project and added the results based on the ENSEMBLES

runs as Appendix D.

- We added a new Appendix C including a new figure containing results of the calculated basin area for a range of model resolutions. It shows that the D8 method is not applicable at model resolutions coarser than 50 km.

- We added a new Appendix B containing detailed information on the run times of mRM.

A point-by-point response can be found below. Comments are in *italic* font; answers are in blue font.

**Reviewer #1**

*1 General remarks*

*The study presents the mRM which specialized in the routing of main channel flow. It is using an adaptive time step and scales well for different resolutions. It is able to upscale river flow networks and parameters to the input data resolution by itself and thus provides a very user friendly tool for the computation of river discharge from different models. The manuscript is very well written and structured. The reported analysis convincingly support the conclusions. Limitations (e.g. missing floodplain processes) are listed together with future plans. For LSMs without a native routing scheme, the mRM is definitely an interesting tool although the apparent limitation of using only total runoff instead of the different components might limit its applicability somewhat. It would be nice if the authors could add a short paragraph about the resolution limit on the coarse side. Large scale earth system models are usually much coarser than 50km and focus on routing the discharge into the correct ocean basin*

*rather than using it for evaluation. Would this tool be applicable at such resolutions as well?*

We would like to thank the reviewer for her/his effort and time to review our manuscript. We appreciated the reviewer's constructive comments that helped to further improve our manuscript. In detail, we now mention in our manuscript that mRM can be used as a model diagnostic on different runoff components of a land-surface model (e.g., surface runoff and subsurface runoff). We have added this on p. 7, l. 10f and also applied it in our application example with REMO, which provides further insights into the partitioning of precipitation into different runoff components (see p. 17, l. 1f). However, the sum of all runoff components constitute the default input for mRM that should be used for model prediction. Following the advice from the reviewer, we added a section in the appendix on the upscaling method of the river network, the D8 method (see Appendix C). Although this method is well established in the community, this section shows that the common limitations of the D8 method on coarser scales are also present in mRM. The basic limitation of the D8 method is that calculated drainage area increases with resolution and large mismatches between calculated and actual drainage area can occur at coarse scales. Improvements of the D8 method at low resolutions have been proposed by Yamazaki et al. (2009), but these are not used in mRM for two reasons. First, large differences between calculated basin areas and observed basin areas only exists in simulations with multiple outlets. If mRM is setup for one catchment, where the entire model domain drains to one outlet, then the calculated basin area is close to the observed basin area and independent of the model resolution. These single-basin setups are frequently used for parameter estimation to minimize the computational demand. Second, this problem is negligible for continental scale simulations at resolutions of quarter degree or less. mRM allows to conduct such simulations even if the input (i.e., gridded runoff field) is provided at 1 degree without any modification of the input data. The provided input runoff is internally scaled to the model resolution. We have added the new Figure C1 in Appendix C to discuss this issue. These new results also answer the last question of the reviewer. Yes, mRM can be used to route water into the

correct ocean basin (see p. 7, l. 5). It is currently setup for the whole globe and first results have been presented by Oldrich Rakovec at the EGU-2019 general assembly (https://meetingorganizer.copernicus.org/EGU2019/EGU2019-13125.pdf).

*2 Specific remarks*

*P2L12: I don't understand this sentence. I guess you mean you provide a framework for those LSMs without a native river routing scheme to compute river discharge and compare it to observations? Please rephrase.*

We rephrased this sentence to "The main goal of this study is to provide LSMs, that do not include a river routing scheme, with a framework to compute streamflow for comparison against observations. The distinctive property of this framework is that the spatial resolution can be easily changed by the user without any modification of the model setup." (see p. 2, L. 12ff).

*P17L1: Would this mean that using mRM with a LSM that generates multiple runoff components, e.g. fast runoff, baseflow..., that mRM would have to be applied separately for each of them with a component specific celerity? Please clarify.*

No, mRM would in general be applied to the sum of all runoff components based on the assumption that all of these enter the river in the same grid cell. However, it is possible to apply mRM to different components individually which might be an interesting diagnostic. We have conducted these runs for the REMO model, which revealed that surface runoff makes up almost all runoff generated by REMO (see p. 16, l. 34ff). We also provide an answer to the question by the reviewer in the methods section (see p. 7, l. 8ff).

*P17L1: REMO does separate runoff into two components: surface runoff and drainage. How are they treated for this study? Are they just combined to total runoff?*

We updated the REMO analysis to the runs provided within the EURO-CORDEX project (see Section 3.4). In general, mRM is applied to the sum of all runoff components (see answer above). We also applied mRM to surface runoff component of REMO, which revealed that surface runoff makes up almost all runoff of REMO.

*P17L25: The statement seems a bit unfair. As just said REMO does compute different runoff components and (according to the output variable list) they should be available from the ENSEMBLES database. Also, ENSEMBLES is not exactly the newest project out there. Why not using data from the EURO-CORDEX Project which would allow to draw conclusions about REMO that would be much more up to date. ESGF has the variables total runoff (mrro) and surface runoff (mrros) available (drainage (mross) would just be mrro – mrros for this model). Having said this, I like to stress that this does not compromise the manuscript in any way as the REMO analysis is just an example for the functionality of mRM. Thus, no changes are necessary here.*

Following your advise, we have updated the REMO analysis using the runs provided by the EURO-CORDEX in favor of the ENSEMBLES runs. We also apply mRM only to the surface runoff component (see answer above). We are deeply convinced that runoff generation in land-surface models needs to be improved. The high fraction of surface runoff in REMO and the fast response of surface runoff is rather unrealistic at the resolution that REMO is applied. We have seen similar behaviour in land-surface models. This is, in our opinion, the main reason why impact models are still used in climate change studies like ISI-MIP. However, we have rephrased this sentence to make clear that REMO is just one example of this general problem of runoff generation in land-surface models (see p. 18 l. 25ff).

*P21L26: Is this a left-over from an earlier submission? A bit early to thank for constructive comments before knowing what you get ;) . Still, I appreciate the attitude. Btw, at least I was contacted by a different editor and also I cannot see Paul Dirmeyer in the Editorial Board of GMD.*

We apologize for this blunder on our side. We are thankful to the handling editor Jeffrey Neal, Thomas Riddick, and yourself for taking the time to consider our work

and providing constructive feedback. Of course, we have corrected this mistake (see p. 26, l. 6ff).

*3 Technical remarks*

*P1L12: are they really identical? I guess you mean similar, right?*

Yes, they are similar. We corrected it (see p. 1, l. 12).

*P1L14: everything is basically comparable. Do you mean similar again?*

This sentence has been removed because the EURO-CORDEX simulation did not provide comparable simulations across scales.

**Reviewer #2 – Thomas Riddick**

*1 General comments*

*This paper presents a model for simulating river discharge across a range of resolutions. This will likely be a very useful tool for a range of applications and its high degree of flexibility is a particular strong point. I generally find the model to be well presented and the validation provided on two different datasets to be thorough. The quality of the figures given is good and the quality of the writing generally high though there a few notable lapses (see technical corrections for those I have noticed). My only concern is a lack of discussion of the upscaling of river directions themselves; particularly to the coarser scales. Upscaling D8 river directions can produce errors in routing that leads to water entering the wrong catchment. Did the authors encounter such problems? Would they expect to encounter them for river systems outside Europe? Were other upscaling algorithms (e.g. Yamazaki et al. 2009 "Deriving a global river network map and its sub-grid topographic characteristics from a fine-resolution flow direction map")*

*considered at any point? Even if uncommon such routing errors would have a major impact on model performance so are worthy of some consideration/discussion.*

We thank you, Thomas, for your time and effort to provide feedback on our manuscript. We are happy to hear that you like our work. We apologize for notable lapses in our writing and have conducted a thorough spell checking again. Following your advice, we have added Appendix C including a new figure that present the impact of model resolution on calculated drainage area for major European river basins. We are aware of the deficiencies of the D8 method that are also becoming apparent in mRM at resolutions coarser than 50 km. We have, however, not implemented the augmentations suggested by Yamazaki et al. (2009). The reason is that these deficiencies are only present at coarse resolutions (e.g., larger than 50 km for investigated basins). mRM allows to choose a higher routing resolution in a configuration file, e.g. 0.25 degree, irrespective of the resolution of the provided input (i.e., gridded runoff field). The user does not need to modify the runoff input, neither on higher nor on lower resolution (e.g., 1 degree). mRM would, in this case, automatically downscale the runoff from the input resolution (1 degree) to the model resolution (0.25 degree) and circumvent the problem of D8 at coarse resolutions.

*2 Specific comments*

*Page 3 Line 2 – Minimization of computational demands is mentioned here but unless I missed it no indication of the actual computational demand of this scheme is mentioned here. Ballpark figures for a one or two resolutions and grid sizes would be useful for comparison purposes – how long does a simulation of X timestep/days/years take on machine Y? Is parallelisation possible?*

We have added Appendix B with detailed run times of mRM. Please note that mRM is "data-hungry" software in the sense that a substantial part of the run time is taken up for reading and writing files. This implies that run times can be substantially larger if I/O is slow. Parallelisation is a non-trivial task because the calculations need to be carried out along the river. In other words, it is not possible to calculate the flow at downstream cells before head waters have been calculated. Maren Kaluza, a Phd student in our group, is currently parallelising the code for its application at high-performance compute cluster (i.e., on the order of thousands compute cores used). She presented first promising results at the EGU 2019 (https://meetingorganizer.copernicus.org/EGU2019/EGU2019-8129-1.pdf). We mention parallelisation as a next development step in the conclusions (see p. 21, l. 16f).

*Page 4 Line 7 – What does "center grid" mean here? Is this supposed to mean "grid center"?*

Yes, this is supposed to mean "grid center". We corrected it (see p. 4, l. 10).

*Page 9 Line 19 – "exhibits limited impact to change of epsilon and gamma." – I am not quite sure what is meant here. Please review this sentence*

We have rephrased this sentence to: "almost independent of the choice of $\epsilon$ and $\gamma$" (see p. 11, l. 5).

*Page 15 Line 31 – This sentence isn't clear. I assume it is meant that the model can also handle rotated grid as well as regular latitude longitude grids. Please clarify. Also, how about other grid e.g. a triangular grid?*

We have moved this sentence to the methods section 2.4 and rephrased it to: "Note that mRM can also handle rotated grids, if the high-resolution digital elevation model is provided on a rotated grid. " (see p. 7, l. 6). An implementation of triangular grids is not foreseen.

*Page 18 Line 7 – "It also avoids further computational demand by scaling the generated runoff etc. etc.". I wasn't clear what was meant by this sentence. Please clarify.*

We were referring to the fact that hyper-resolution have a higher computational demand than low resolution simulations. We have removed the words "computational demand by" from this sentence to avoid confusion (see p. 19, l. 9f).

*3 Technical corrections*

We are very impressed by the amounts of technical corrections you provided. We have corrected all of them and apologize that so many were present. We have conducted a thorough spell checking and were able to remove more spelling mistakes. Please see the manuscript with highlighted changes for details. We have only provided answers below if we deviated from your suggestion.

*Page 5 Line 9 – "the equation 5" → "equation 5"*

*Page 5 Line 9 – Incomplete sentence "the sufficiently high model performance" → "the sufficiently high model performance that it gives."*

We have rephrased this to "its sufficiently high model performance and its simplicity" because it is shorter (see p. 5, l. 13).

*Page 5 Line 11 – I suggest "but it is not as available as. . ." → "but it is not as readily available as. . ."*

*Page 6 Line 6 – "dividers" → "divisions"*

We changed these to "fractions" because this fits better here (see p. 6, l. 13).

*Page 7 Line 13 – "along → "on"*

*Page 8 Line 6 – "daily resolution NetCDF files" → "daily resolution in NetCDF files"*

*Page 9 Line 9 – "which was obtained" → "which were obtained"?*

We rephrased to: "The observed values were..." (see p. 9, l. 27).

*Page 9 Line 13 – Invalid grammatical construction "because they allow to investigate" → I suggest to replace it with "because they allow us to investigate"*

We rephrased to: "because the hydrologic cycle can be investigated" (see p. 9, l. 30).

*Page 11 Line 23 – "aTS allows to use different" → "aTS allows us to use different"*

We rephrased to: "...aTS allows the user to choose..." (see p. 12, l. 12).

*Page 12 Line 2 – "very little standard deviations" → "very low standard deviations"*

*Page 12 Line 9 – "which generates the little mismatch" → "which generates the small mismatch"*

*Page 13 Figure 5 Caption – "light gray violins", "dark gray violins". I see only red and blue violins. It seems the color scheme has been updated without updating the caption. Please update this caption.*

*Page 13 Figure 5 Caption – "and have scaled to the same widths" → "and are scaled to the same widths"*

*Page 15 Line 12 – "The aTS parameterization allows to simultaneously" → "The aTS parameterization allows us to simultaneously"*

We rephrased to: "The aTS parametrization allows the user simultaneously scale" (see p. 16, l. 2).

*Page 15 Line 18 – "still little" → "still small"*

*Page 15 Line 18 – "that has the hydrological model used as input to the routing scheme." → "that the hydrological model used as input to the routing scheme has."*

*Page 17 Line 12 – "very little" → "very low"*

**Supplement:**

[revised manuscript text omitted]